# Music Flamingo: Scaling Music Understanding in Audio Language Models

**Sreyan Ghosh**[12*]**, Arushi Goel**[1*]**, Lasha Koroshinadze**[2**]**, Sang-gil Lee**[1]**, Zhifeng Kong**[1]**,
Joao Felipe Santos**[1]**, Ramani Duraiswami**[2]**, Dinesh Manocha**[2]**, Wei Ping**[1]**,
Mohammad Shoeybi**[1]**, Bryan Catanzaro**[1]

NVIDIA, CA, USA[1], University of Maryland, College Park, USA[2]

Correspondence: sreyang@umd.edu, arushig@nvidia.com

Project: `https://research.nvidia.com/labs/adlr/MF/`

## Abstract

We introduce **Music Flamingo**, a novel large audio–language model, designed to advance music (including song) understanding in foundational audio models. While audio–language research has progressed rapidly, music remains challenging due to its dynamic, layered, and information-dense nature. Progress has been further limited by the difficulty of scaling *open* audio understanding models, primarily because of the scarcity of high-quality music data and annotations. As a result, prior models are restricted to producing short, high-level captions, answering only surface-level questions, and showing limited generalization across diverse musical cultures. To address these challenges, we curate MF-Skills, a large-scale dataset labeled through a multi-stage pipeline that yields rich captions and question–answer pairs covering harmony, structure, timbre, lyrics, and cultural context. We fine-tune an enhanced Audio Flamingo 3 backbone on MF-Skills and further strengthen multiple skills relevant to music understanding. To improve the model's reasoning abilities, we introduce a post-training recipe: we first cold-start with MF-Think, a novel chain-of-thought dataset grounded in music theory, followed by GRPO-based reinforcement learning with custom rewards. Music Flamingo achieves state-of-the-art results across 10+ benchmarks for music understanding and reasoning, establishing itself as a generalist and musically intelligent audio–language model. Beyond strong empirical results, Music Flamingo sets a new standard for advanced music understanding by demonstrating how models can move from surface-level recognition towards layered, human-like perception of songs. We believe this work provides both a benchmark and a foundation for the community to build the next generation of models that engage with music as meaningfully as humans do.[1]

*"Without music, life would be a mistake."* — Friedrich Nietzsche

## 1 Introduction

Audio – including speech, environmental sounds, and music – is central to human perception and interaction. It enables us to converse, perceive our surroundings, express emotions, interpret multimedia, and engage with cultural artifacts. Among these, music is particularly significant: the creation, sharing, discovery, and understanding of music are daily activities for billions worldwide. Recent progress in Audio–Language Models (ALMs) has extended language models into the auditory domain, enabling impressive advances in speech and sound understanding. Yet, music remains fundamentally distinct from other forms of audio. Core musical attributes such as key, tempo, harmony, instrumentation, and vocal styles are not present in non-musical audio and require specialized reasoning. Moreover, tasks adapted from speech and sound (*e.g.*, captioning, transcription, retrieval) demand unique treatment when applied to music. To date, no model has achieved music understanding on par with the multi-modal breakthroughs seen in vision or speech. Improved music understanding

---

[1][*]Equally contributed and led the project. Names randomly ordered.[**] Significant technical contribution.

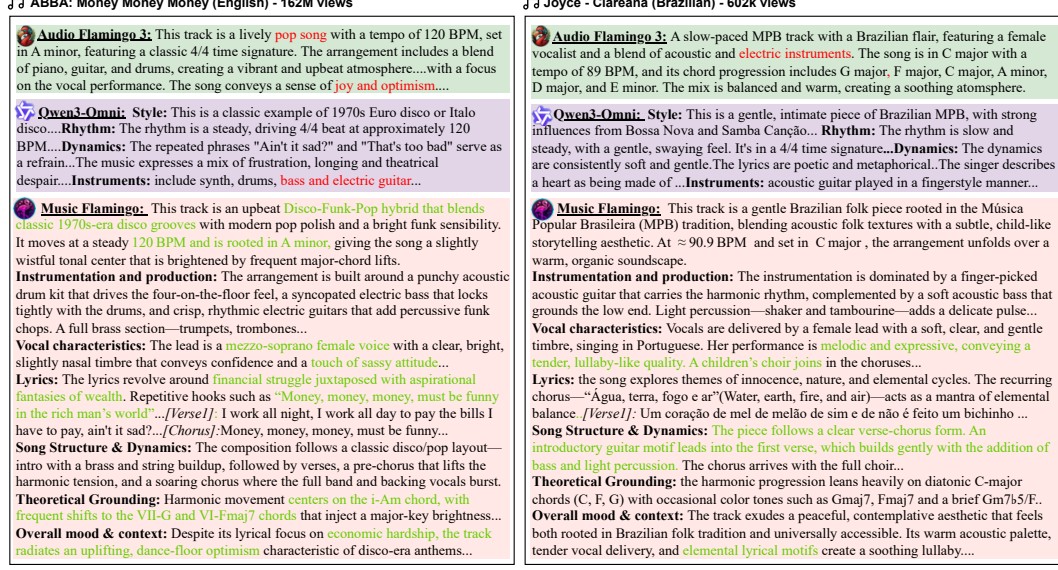

Figure 1: Comparison of captions for two diverse, full-length, in-the-wild songs by **Music Flamingo** and other frontier models. Prior models, such as AF3, tend to output short, surface-level descriptions (e.g., broad genre, tempo, or instrumentation), while Qwen3-Omni offers isolated observations without forming a coherent musical narrative. In contrast, Music Flamingo produces detailed, multi-layered captions that integrate theory-aware analysis with performance context. It links surface attributes (tempo, key, etc.) to mid-level structures (chord progressions, vocal phrasing, etc) and higher-level dimensions (lyrical meaning, emotional trajectory, etc.). This ability to connect one aspect of music to another results in richer, more holistic captions that resemble how trained musicians describe songs. Detailed expert analysis in Appendix E and F.

would unlock richer applications in creation, recommendation, cross-cultural analysis, education, and interactive systems, enabling models to engage with music as deeply as humans do.

Despite advances in scaling LALMs (Goel et al., 2025; Chu et al., 2024; KimiTeam et al., 2025; Tang et al., 2024; Bertin-Mahieux et al., 2011), effective music understanding remains an open challenge (hereafter, we use "music" to refer broadly to both instrumental pieces and songs). Current frontier LALMs, when captioning even widely recognizable tracks, often produce short and generic descriptions, misidentify surface-level attributes such as tempo or key, and sometimes rely on text-derived knowledge rather than genuine auditory analysis (Gemini team, 2025). We argue this stems largely from data: most available music–caption pairs originate from early datasets like MusicCaps (Agostinelli et al., 2023b), and subsequent datasets inherit its stylistic limitations of short, surface-level summaries, limitations of short, surface-level summaries that omit bar/time localization, harmonic and formal structure, vocal/lyric grounding, and cultural context, and often with a narrow focus on instrumental-only snippets. This prevents models from learning the layered nature of music, spanning surface attributes (tempo, key, timbre), mid-level structures (chord progressions, rhythm, phrasing), and higher-level dimensions (lyrics, emotional arcs, cultural context). Architecturally, training practices for most music LLMs or captioners still constrain holistic learning – for example, the use of encoders (like CLAP (Elizalde et al., 2022)) that do not capture spoken content or low-level features like pitch in their representations (see our study in Appendix G), thereby constraining learning of vocal timbre, lyrical alignment, and expressive nuances in songs. We contend that even a task as basic as music captioning, when re-imagined beyond surface-level summaries, is inherently explorative and compositional: a musically informed description requires reasoning through multiple layers of structure and meaning, and admits not one single answer but a spectrum of valid interpretations shaped by theory, perception, and artistry.

**Main Contributions.** In this paper, we introduce **Music Flamingo**, a new and *open-source* large audio–language model specifically designed to advance music understanding. Unlike speech or environmental sounds, music is inherently *layered, expressive, and structured*, combining surface-level acoustic attributes (tempo, key, timbre) with mid-level organization (harmony, form, rhythm) and higher-level dimensions (lyrics, style, affect, cultural context). Capturing this multi-faceted nature of music requires models that can move beyond surface-level recognition toward reasoning and interpretation more akin to a trained musician.

To build Music Flamingo, we re-imagine the scope of music understanding and recast conventional tasks, such as music captioning and question answering, into comprehensive formulations that demand deliberate, step-by-step reasoning (Fig. 3). To support this reframing, we introduce new strategies for both data curation and model training. First, we present **MF-Skills**, a dataset with 4M+ high-quality samples for training music-understanding models. Unlike prior corpora dominated by short, instrumental snippets, MF-Skills scales to long, multicultural full-length songs with vocals drawn from diverse sources. We propose a multi-step labeling pipeline that yields detailed, multi-aspect, *layered* captions – capturing harmony, structure, timbre, lyrics, and cultural context – designed to elicit musician-level reasoning. Beyond captions, MF-Skills includes carefully curated question–answer pairs that move past simple instrument identification toward tasks requiring temporal understanding, harmonic analysis, lyrical grounding, and other skills. On the modeling side, we first identify core limitations in Audio Flamingo 3 and continue-pre-training it to build a stronger backbone by fine-tuning it on multilingual, multi-speaker ASR and extended audio reasoning datasets before specializing it for music. Next, we propose a post-training stage specifically designed to enhance reasoning. For this stage, we further introduce **MF-Think**, a dataset of 300K chain-of-thought examples grounded in music theory, which we use for cold-start reasoning training. Finally, we apply GRPO-based reinforcement learning with custom rewards, enabling explicit step-by-step musical reasoning. In summary, our contributions are:

- We propose **Music Flamingo**, a new LALM for advancing music understanding. We re-imagine conventional music tasks (*e.g.*, captioning, QA) as reasoning-centric formulations and introduce novel training strategies tailored to these tasks.
- To support training, we release **MF-Skills** and **MF-Think**, two large-scale datasets containing music–caption and music–QA pairs designed to promote deliberate reasoning. Unlike prior datasets limited to short instrumental clips, ours include full-length, multi-cultural songs with detailed, multi-aspect annotations.
- Music Flamingo achieves state-of-the-art results on 12 music understanding and reasoning benchmarks. Beyond academic benchmarks, expert evaluations show its outputs are more accurate and preferred by trained musicians than existing models.
- We have released code, training recipes, and our new datasets under an appropriate research-only license at `https://research.nvidia.com/labs/adlr/MF/`.

## 2  RELATED WORK

**Multimodal audio–language modeling.** The rapid progress of LLMs has accelerated the development of multimodal LLMs (MLLMs) capable of understanding and reasoning across diverse modalities, including audio. Within this space, ALMs focus specifically on reasoning over auditory inputs such as speech, sounds, and music. Architecturally, ALMs generally follow two paradigms: (i) *Encoder-only ALMs*, which learn a joint embedding space for audio and text, enabling tasks like cross-modal retrieval. Representative models include CLAP (Elizalde et al., 2022), Wav2CLIP (Wu et al., 2021), and AudioCLIP (Guzhov et al., 2021). (ii) *Encoder–decoder ALMs* (often called Large Audio–Language Models, LALMs), which augment decoder-only LLMs with audio encoders. Notable examples include LTU (Gong et al., 2023b), LTU-AS (Gong et al., 2023a), SALMONN (Tang et al., 2024), Pengi (Deshmukh et al., 2023), Audio Flamingo (Kong et al., 2024), Audio Flamingo 2 (Ghosh et al., 2025), Audio Flamingo 3 (Goel et al., 2025), AudioGPT (Huang et al., 2023), GAMA (Ghosh et al., 2024), Qwen-Audio (Chu et al., 2023), and Qwen2-Audio (Chu et al., 2024). There has also been a surge of LALMs that specifically focus on music, including Mu-LLaMA (Liu et al., 2024a), MusiLingo (Deng et al., 2024), M2UGen (Liu et al., 2024b), SALMONN (Tang et al., 2024), and LLARK (Gardner et al., 2024). These LALMs have substantially advanced core audio understanding tasks such as automatic speech recognition (ASR) (Radford et al., 2022), audio captioning (Kim et al., 2019), and acoustic scene classification (Chen et al., 2022). More importantly, they have enabled *open-ended audio question answering*, which requires both complex auditory reasoning and external world knowledge. While music has often been included as a modality within these models, it has rarely been a central focus.

Scaling music understanding within ALMs has proven particularly difficult. For instance, while the Audio Flamingo series has expanded its training data substantially from version 1 to 3, the music component of training data has increased by only ≈10%, compared to much larger growth in speech and environmental sounds. Similarly, models such as Kimi (KimiTeam et al., 2025) and

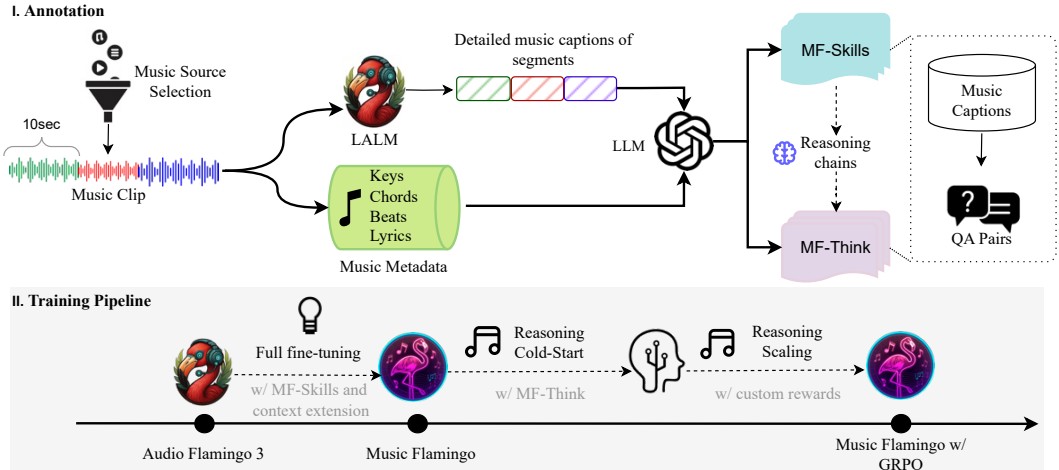

Figure 2: **I. Annotation pipeline** for constructing our proposed datasets from diverse music clips. **II. Training pipeline** of Music Flamingo: we begin by improving Audio Flamingo 3, then perform full fine-tuning on music datasets to build the Music Flamingo foundation model. Finally, the model undergoes reasoning cold-start training followed by GRPO fine-tuning to enable step-by-step reasoning.

Step Audio (Huang et al., 2025) (where training data disclosures exist) show comparable imbalances. Finally, models like LLARK (Gardner et al., 2024) and MU-LLaMA (Liu et al., 2024a) curate music captions and question-answer pairs from existing open-source datasets, which lack diversity and skills. This is due to several challenges: the difficulty of collecting high-quality and culturally diverse music audio (Kumar et al., 2025), curating reliable annotations (Christodoulou et al., 2024), and the reliance of most works on private, proprietary datasets (Agnew et al., 2024). Large labs often construct in-house collections of lyrics and metadata by scraping online lyric repositories, translations, and song databases (Ahmed et al., 2025). Models such as Jukebox and Neural Melody Reconstruction exemplify this paradigm. Finally, as noted earlier, most publicly available datasets emphasize short instrumental clips, with very limited coverage of full-length songs containing vocals, hindering a comprehensive understanding of music (Kumar et al., 2025).

**Music information retrieval and captioning.** Beyond LALMs, music understanding has a long history in Music Information Retrieval (MIR), encompassing retrieval, classification, and captioning. Foundational tasks such as key detection (Chai & Vercoe, 2005), chord recognition (Sheh & Ellis, 2003), and tempo estimation (Scheirer, 1998) have been extensively studied, largely in instrumental music. Lyrics transcription has also been explored, posing a more difficult challenge than ASR due to overlapping vocals, diverse singing styles, and background instrumentation (Mesaros & Virtanen, 2010). As discussed earlier, music captioning has been studied in analogy to audio event captioning, but typically produces short, high-level semantic descriptions rather than layered, theory-aware accounts. Importantly, improved captioning not only benefits downstream music understanding but also supports the training of generative music models by providing high-quality text supervision for in-the-wild audio (Agostinelli et al., 2023b). This connection has recently been emphasized in both standalone music modeling efforts and in broader video generation systems (Chen et al., 2025).

## 3 METHODOLOGY

To build Music Flamingo, we first curate high-quality songs, followed by labeling them and finally fine-tuning the model on the curated data. Music Flamingo is a specialized music understanding model built by fine-tuning a version of Audio Flamingo 3, specifically with high-quality data to close the gap on skills and tasks crucial for music understanding. Finally, the model is fine-tuned using reinforcement learning to enable step-by-step music reasoning.

### 3.1 IMPROVED AUDIO FLAMINGO 3 BASELINE

**Data.** We first strengthen Audio Flamingo 3 to serve as the backbone for Music Flamingo. Unlike instrumental-only music, songs contain vocals that contribute not only lyrics but also timbre, style,

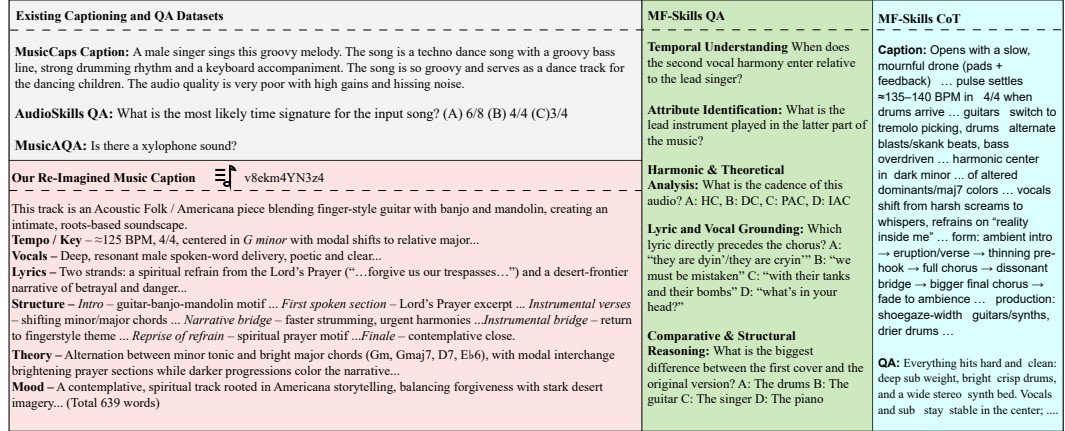

| Existing Captioning and QA Datasets | MF-Skills QA | MF-Skills CoT |
|---|---|---|
| **MusicCaps Caption:** A male singer sings this groovy melody. The song is a techno dance song with a groovy bass line, strong drumming rhythm and a keyboard accompaniment. The song is so groovy and serves as a dance track for the dancing children. The audio quality is very poor with high gains and hissing noise.

**AudioSkills QA:** What is the most likely time signature for the input song? (A) 6/8 (B) 4/4 (C)3/4

**MusicAQA:** Is there a xylophone sound? | **Temporal Understanding** When does the second vocal harmony enter relative to the lead singer?

**Attribute Identification:** What is the lead instrument played in the latter part of the music?

**Harmonic & Theoretical Analysis:** What is the cadence of this audio? A: HC, B: DC, C: PAC, D: IAC

**Lyric and Vocal Grounding:** Which lyric directly precedes the chorus? A: "they are dyin'/they are cryin'" B: "we must be mistaken" C: "with their tanks and their bombs" D: "what's in your head?"

**Comparative & Structural Reasoning:** What is the biggest difference between the first cover and the original version? A: The drums B: The guitar C: The singer D: The piano | **Caption:** Opens with a slow, mournful drone (pads + feedback) … pulse settles ≈135–140 BPM in 4/4 when drums arrive … guitars switch to tremolo picking, drums alternate blasts/skank beats, bass overdriven … harmonic center in dark minor … of altered dominants/maj7 colors … vocals shift from harsh screams to whispers, refrains on "reality inside me" … form: ambient intro → eruption/verse → thinning pre-hook → full chorus → dissonant bridge → bigger final chorus → fade to ambience … production: shoegaze-width guitars/synths, drier drums …

**QA:** Everything hits hard and clean: deep sub weight, bright crisp drums, and a wide stereo synth bed. Vocals and sub stay stable in the center; |
| **Our Re-Imagined Music Caption** ♫ v8ekm4YN3z4 | | |
| This track is an Acoustic Folk / Americana piece blending finger-style guitar with banjo and mandolin, creating an intimate, roots-based soundscape.
**Tempo / Key** – ≈125 BPM, 4/4, centered in *G minor* with modal shifts to relative major...
**Vocals** – Deep, resonant male spoken-word delivery, poetic and clear...
**Lyrics** – Two strands: a spiritual refrain from the Lord's Prayer ("…forgive us our trespasses…") and a desert-frontier narrative of betrayal and danger...
**Structure** – *Intro* – guitar-banjo-mandolin motif … *First spoken section* – Lord's Prayer excerpt … *Instrumental verses* – shifting minor/major chords … *Narrative bridge* – faster strumming, urgent harmonies …*Instrumental bridge* – return to fingerstyle theme … *Reprise of refrain* – spiritual prayer motif …*Finale* – contemplative close.
**Theory** – Alternation between minor tonic and bright major chords (Gm, Gmaj7, D7, E♭6), with modal interchange brightening prayer sections while darker progressions color the narrative...
**Mood** – A contemplative, spiritual track rooted in Americana storytelling, balancing forgiveness with stark desert imagery... (Total 639 words) | | |

Figure 3: Examples from MF-Skills Caption , MF-Skills QA , and MF-Think . We emphasize that our re-imagined captions are denser, more informative, and designed to require deliberate reasoning to generate. Additional examples are provided in Appendix G.1.

and expressive variation. Capturing these elements requires stronger spoken language understanding than prior baselines. Thus, in addition to the data used for AF3 training, we add the following to the mix: 1) Across all fine-tuning stages (1–3), we incorporate large-scale multilingual ASR data (sources Emilia dataset (He et al., 2024), CoVoST (Wang et al., 2020), MUST (Qin et al., 2025), Amazon-SIFT (Pandey et al., 2025); details in Appendix 5) to better capture global vocal diversity, 2) In stage 3, we add multi-talker ASR data, including CHIME (Watanabe et al., 2020; Cornell et al., 2023), Switchboard (Godfrey et al., 1992) and ALI meeting (Yu et al., 2022), enabling the model to parse turn-taking and overlapping voices, which is critical for understanding duets and ensemble singing and, 3) We expand the data mix with speech-centric skills, including phoneme recognition and lyrics transcription, improving alignment between vocal content and musical context.

**Training Pipeline.** We adopt the training paradigm introduced in Audio Flamingo 3 (Goel et al., 2025) fine-tune the model on the diverse set of speech data described above. The resulting fine-tuned model then serves as the foundation for developing the music-focused foundational model.

## 3.2 BUILDING FOUNDATIONAL MUSIC UNDERSTANDING

**MF-Skills.** Prior captioning datasets mostly provide surface-level summaries, while existing QA datasets are dominated by simple classification tasks (*e.g.*, instrument or tempo detection). Even large-scale skill-focused datasets such as AudioSkills focus primarily on sounds and speech for their diverse skill-specific QAs, with music data reduced to basic information extraction. To mitigate this gap, we design **MF-Skills** and capture the layered nature of music to train models for deliberate reasoning. Figure 2 illustrates our data curation pipeline, and Figure 3 provides examples from our curated dataset.

We begin by collecting full-length songs from diverse cultures (∼3M in total), as shown in Figure 4, thereby moving beyond the short, Western instrumental clips that dominate prior datasets. As shown in Figure 2, our pipeline consists of four stages: 1) **Initial caption synthesis:** Generate short, surface-level captions for 30s segments using frontier music models to minimize hallucinations, 2) **Metadata extraction:** We apply conventional MIR tools, including

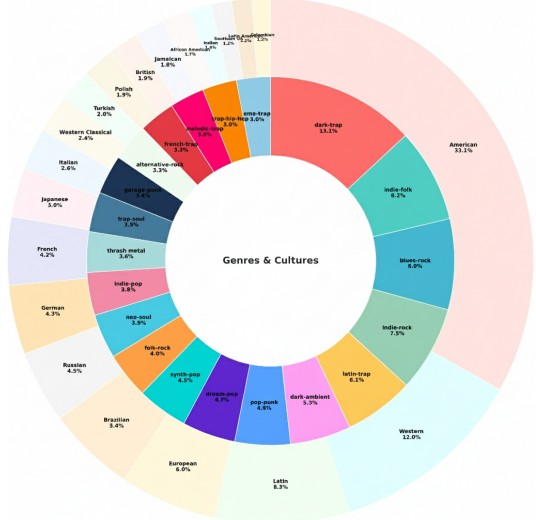

Figure 4: Genres (inner circle) & Cultures (outer circle) distribution of songs.

`madmom` (Böck et al., 2016) (beat), `essentia` (Bogdanov et al., 2013) (key), `Chordino` (Mauch & Dixon, 2010) (chords), `Parakeet` (NVIDIA, 2025) (lyrics)–to provide reliable low-level attributes and, 3) **Caption & QA creation:** Using metadata and initial captions, we prompt an LLM (with music-theory grounding) to produce detailed, multi-aspect captions covering several aspects including: a) low-level information (tempo, BPM, keys), b) instrumentation & production, c) lyrics, and lyrical themes (including structural segmentations such as verses, choruses, and bridges), d) song structure & dynamics e) theoretical insight (*e.g.*, chord transitions and harmonic movements) and f) overall mood & context. Our final captions have an average of 451.65 words. For QA, we analyze skill gaps in AF3 using benchmarks such as MMAU (Sakshi et al., 2024), MMAU-Pro (Kumar et al., 2025), MuChoMusic (Weck et al., 2024), MusicCaps (Agostinelli et al., 2023b), MusicQA (Li et al., 2022), and NSynth (Engel et al., 2017), then generate novel QA targeting five skills: a) Temporal understanding, b) Attribute identification, c) Harmonic & theoretical analysis, d) Lyric and vocal grounding, e) Comparative and structural reasoning. This approach also mitigates distribution gaps, *e.g.* instrument identification in complex, multi-layered songs rather than isolated clips, and culture–skills gaps, *e.g.* identifying ragas in Indian music or polyrhythms in African drumming, which prior datasets neglect as they were dominated by Western music. 4) **Quality filtering:** A frontier MLLM is used to verify and retain only high-quality captions and QAs. The final dataset contains ∼5.2M examples (∼3.4M captions and ∼1.8M QAs).

Beyond curating new data, we also refine existing music datasets such as MSD (Bertin-Mahieux et al., 2011), Music4All (Geiger et al., 2025), and the music subset of AudioSkills-XL (Goel et al., 2025). We (a) rewrite captions to add lyrical themes, vocal attributes, and correct mislabels of tempo, key, and timbre using metadata, and (b) reframe MCQ-style questions to reduce language priors and guessing, an issue highlighted in recent benchmarks (MMAU-Pro (Kumar et al., 2025), RUListening (Zang et al., 2025)). We cluster Q&As by trait and rephrase them with metadata to require genuine auditory perception. We provide examples below:

> ***Existing caption (from AudioSkills-MSD):*** This is an upbeat 1980s pop-rock track with a danceable 4/4 beat around 140 BPM, featuring bright guitar riffs and melodic synthesizers. The song carries an energetic and catchy feel, blending indie rock elements with disco-inspired rhythms, typical of early 80s production.
>
> ***Our modified caption:*** This upbeat 1980s pop-rock track in B minor rides a danceable 4/4 beat around 120 BPM, driven by bright electric guitars, shimmering synths, and lively drums. Its catchy, energetic melody blends indie and disco influences, ⋯ The lyrics add a layer of intimate storytelling, weaving lines about ⋯ moments of fleeting connection and dreamy recollection ("I close my eyes and count to ten ⋯ we were strangers a moment ago").
>
> ***Existing QA (from AudioSkills-MSD):*** What genre does this track? Choose one among the following options: (A) Jazz (B) Classical (C) Rock (D) Spoken Word ("Spoken Word" stands out as a unique option among all other options, which are music genres)
>
> ***Additional plausible distractors options added to modified QA:*** (E) Audiobook narration excerpt (F) Rap a cappella (no beat) (G) Podcast monologue intro (H) Documentary voice-over bed (I) Theatrical monologue with ambiance (J) Spokenword/poetryy

**Training Methodology.** We begin with the improved base model derived from re-training Audio Flamingo 3, as outlined in Section 3.1. This model is subsequently fine-tuned on the proposed MF-Skills dataset, the improved QA datasets described above, and other music datasets derived from the training mix of Audio Flamingo 3 (Goel et al., 2025). Inspired by our study in Appendix G, we also incorporate data for learning low-level music properties, such as chords, keys, and BPM. Dataset details in Appendix C

We also encounter two primary limitations in training the model. First, the Audio Flamingo 3 backbone supports a maximum context length of 8,192 tokens and ∼10 minutes of audio, whereas our curated datasets predominantly contain much longer captions and full-length songs up to 20 minutes. We extend the context length to ∼24k tokens, and adopt fully sharded training to handle the increased memory requirements. Second, music understanding requires fine-grained temporal perception, including chord progressions, tempo, key changes, and vocal dynamics. To capture these transitions, we incorporate time-aware representations into the audio encoder outputs before feeding tokens into the LLM. Specifically, we employ Rotary Time Embeddings (RoTE) (Goel et al., 2024),

which define the rotation angle $\theta$ using absolute timestamps rather than token indices. Unlike standard RoPE, where the rotation angle $\theta$ depends on the token index $i$ as $\theta \leftarrow -i \cdot 2\pi$, RoTE defines $\theta$ using the token's absolute timestamp $\tau_i$: $\theta \leftarrow -\tau_i \cdot 2\pi$. For audio tokens produced at a fixed stride of 40ms (Radford et al., 2022; Goel et al., 2025), we interpolate discrete time positions $\tau_i$ and feed them into the RoTE module to obtain lightweight, temporally grounded representations.

### 3.3 POST-TRAINING WITH REINFORCEMENT LEARNING

While prior music understanding tasks rarely required reasoning, our formulation explicitly demands it. For example, generating a caption in Figure 3 requires the model to progressively connect surface properties (tempo, key) with higher-level structures (harmony, form, production, lyrics) and then articulate them as a coherent musical narrative—a process that is non-trivial even for trained musicians. To enable this, we introduce a post-training stage beyond large-scale SFT that strengthens Music Flamingo's reasoning abilities. First, we construct **MF-Think**, a high-quality Chain-of-Thought (CoT) dataset used for cold-start reasoning. We then fine-tune with MF-Think before applying GRPO with custom-designed rewards, encouraging explicit step-by-step reasoning.

**MF-Think.** We begin with a diverse, high-quality subset of MF-Skills. Since not all QAs demand deep reasoning, we sub-sample the most challenging examples by prompting `gpt-oss-120b` with both the audio and QA (prompt in Appendix H). **CoT Generation.** For each selected QA or caption, we prompt `gpt-oss-120b` with metadata from MF-Skills to generate long, theory-grounded reasoning chains. Prompts include constraints on length, grounding to music theory, and exemplar demonstrations. **Quality Filtering.** Each reasoning chain is segmented into smaller steps, which are fact-checked using our post-SFT MF (Yes/No verification against the audio). We rewrite chains with minor errors and discard those with >30% incorrect steps. The final dataset contains $\approx$176k CoT examples, including $\approx$117k QA and $\approx$59k captioning samples, providing a rich foundation for reasoning-enhanced training.

**Supervised Fine-Tuning with MF-Think.** To equip the model with advanced reasoning capabilities, we first perform SFT of the music foundation model on our curated MF-Think dataset. During this stage, we append the prompt: `Output the thinking process in <think> </think> and final answer in <answer> </answer>` to the original questions in the **MF-Think** dataset, to encourage the model to explicitly generate reasoning chains within the `<think> </think>` tags and the final answer within the `<answer> </answer>` tags. This process instills structured reasoning for both the question-answering and caption-generation tasks. This SFT stage acts as an initial warm-up phase, effectively priming the model for subsequent reinforcement learning (RL) fine-tuning.

**GRPO for Music reasoning and understanding.** Building on the advancements in the GRPO algorithm, we adhere to the standard GRPO algorithm to train our model. GRPO obviates the need for an additional value function and uses the average reward of multiple sampled outputs for the same question to estimate the advantage. For each given question $q$, the policy model generates a group of candidate responses $\{o_1, o_2, \dots, o_G\}$ from the old policy $\pi_{\theta_{old}}$ along with their corresponding rewards $\{r_1, r_2, \dots, r_G\}$ which are computed using rule-based reward functions (*e.g.*, format and accuracy). The model $\pi_\theta$ is subsequently optimized using the following objective function:

$$\mathcal{J}(\theta) = \mathbb{E}_{q, \{o_i\}} \left[ \frac{1}{G} \sum_{i=1}^{G} \left( \min\left( \frac{\pi_\theta(o_i|q)}{\pi_{\theta_{old}}(o_i|q)} A_i, \text{clip}\left( \frac{\pi_\theta(o_i|q)}{\pi_{\theta_{old}}(o_i|q)}, 1-\epsilon, 1+\epsilon \right) A_i \right) - \beta D_{KL}(\pi_\theta \| \pi_{ref}) \right) \right] \quad (1)$$

where $\epsilon$ is the clipping range of the importance sampling ratio, $\beta$ is the regularization strength of the KL-penalty term that encourages the learned policy to stay close to the reference policy, and $G$ is the group size, *i.e.*, the number of candidate responses (samples) the policy generates for each input question, which is set to 5 in our experiments. To stabilize training, the sampled rewards are normalized to compute the advantages $A_i$ as: $\frac{r_i - \text{mean}(\{r_1, r_2, \dots, r_G\})}{\text{std}(\{r_1, r_2, \dots, r_G\})}$ Next, we discuss the custom reward functions we designed for GRPO training, which play a pivotal role in optimization.

**Format Reward.** In order to encourage the model to generate outputs that adhere to the prescribed response format, we use the standard regex-based format reward (DeepSeek-AI, 2025). Specifically, the model is instructed to produce reasoning traces enclosed within `<think> </think>` tags,

followed by the final answer enclosed within `<answer> </answer>` tags. If the output strictly follows the required tag structure, the model gets a reward of 1 else 0. This binary reward function ensures that the model learns to consistently produce well-structured responses.

**Accuracy Reward.** For question-answering (QA) tasks, we employ the *accuracy reward* to encourage the model to generate accurate final answers. Given a question with the corresponding ground-truth answer, the model generates a candidate output $o_i$, where the final answer is extracted from within the `<answer> </answer>` tags. The accuracy reward directly matches the normalized predicted and ground truth answers, enforcing strict answer correctness.

**Structured Thinking Reward.** For caption generation tasks, the standard accuracy reward cannot be directly applied due to the long and open-ended nature of the generated captions. To address this, we design a custom reward function that evaluates generated captions against structured ground-truth metadata. To achieve this, we first generate ground-truth structured metadata as shown below using `gpt-oss-120b` (OpenAI, 2025) for the subset of the captions in the MF-Skills dataset as follows:

> {"**Genre**": Americana, "**BPM**": 125, "**Key**": G minor, "**Meter**": 4/4, "**Structure**": Intro, Verse, Verse, Bridge, Solo, Chorus, Outro, "**Instruments**": fingerstyle acoustic guitar, banjo, mandolin, spoken-word voice, "**Vocal Character**": male spoken-word, deep resonant timbre, clear/deliberate, light reverb, "**Lyric Themes**": forgiveness, humility, spiritual prayer, desert frontier, betrayal, outlaw narrative, "**Theory**": G minor center; modal interchange with relative major; F#aug → Eb6 → D7 resolution; Gmaj7/G7 brighten prayer sections, "**Mix Notes**": high-fidelity organic; wide natural stereo panning; minimal reverb; warm, clear, light compression; close-mic intimacy, "**Dynamics**": bridge increases harmonic rhythm/urgency.}

The structured thinking reward function computes a string match for each answer in the category of the structured ground-truth metadata (*e.g.* Genre, Subgenre, BPM etc.) and the generated caption. The total reward is obtained by normalizing the number of matching words by the total number of metadata categories.

The overall reward function used in GRPO training integrates the format reward with the accuracy reward for the data with question-answer subset, while for the caption subset of data, it combines the format reward with the structured reasoning reward.

## 4 EXPERIMENTS

**Experimental Setup.** We train Music Flamingo on 128 NVIDIA A100 GPUs (80GB). Details on batch size, learning rates, and optimizers for each stage of training are in Appendix D.

**Baselines.** We evaluate our model against recent SOTA LALMs, including GAMA (Ghosh et al., 2024), Audio Flamingo (Kong et al., 2024), Audio Flamingo 2 (Ghosh et al., 2025), Audio Flamingo 3 (Goel et al., 2025), Qwen-Audio (Chu et al., 2023), Qwen2-Audio (Chu et al., 2024), Qwen2-Audio-Instruct, Qwen2.5-Omni (Xu et al., 2025), R1-AQA (Li et al., 2025a), Pengi (Deshmukh et al., 2023), Phi-4-mm (Abouelenin et al., 2025), Baichun Audio (Li et al., 2025b), Step-Audio-Chat (Huang et al., 2025), LTU (Gong et al., 2023b), LTU-AS (Gong et al., 2023a), SALMONN (Tang et al., 2024), AudioGPT (Huang et al., 2023), and Gemini (2.0 Flash, 1.5 Pro, 2.5 Flash and 2.5 Pro) (Team et al., 2023), as well as GPT-4o-audio (Hurst et al., 2024). For Table 1, we only compare against open LALMs. All results reported in the tables correspond to the best-performing model.

**Evaluation Datasets.** We evaluate AF3 across a broad set of benchmarks spanning music information retrieval (MIR), question answering, lyrics transcription, reasoning, and our proposed dataset **Song-Caps**. SongCaps consists of 1,000 culturally diverse songs curated to assess captioning capabilities across multiple dimensions (see Section 3.2). Rather than relying on lexical overlap metrics, we evaluate captions using human-expert judgments and LLM-as-a-judge assessments. For MIR, we use NSynth (Source and Instrument) (Engel et al., 2017), MusicCaps (Agostinelli et al., 2023a), Medley-Solos-DB (instrument classification) (Lostanlen et al., 2019), and GTZAN (genre classification) (Tzanetakis & Cook, 2002). For QA and reasoning, we include MusicAVQA (Li et al., 2022), Music Instruct (Deng et al., 2024), MMAU (v05.15.25) (Sakshi et al., 2024), MMAU-Pro (Kumar et al., 2025), MuChoMusic (perceptual version) (Zang et al., 2025; Weck et al., 2024), and MMAR (Ma et al., 2025). For lyrics transcription, we evaluate on Opencpop (Wang et al., 2022) – a dataset for chinese songs and MUSDB18 Lyrics (Rafii et al., 2019) – a dataset for English songs. *We*

Table 1: Comparison of Music Flamingo (w/ GRPO) with other LALMs on various benchmarks (WER ↓ (Word Error Rate), ACC ↑ (Accuracy), Score (1-10) ↑ and GPT5 ↑ (GPT evaluation)). We report scores for only the top-performing prior LALM. We highlight closed source, open weights, and open source models.

| Task | Dataset | Model | Metrics | Results |
|------|---------|-------|---------|---------|
| **Music QA and Reasoning** | **MMAU (Music)** *full-test \| test-mini* | Audio Flamingo 3 Music Flamingo | ACC ↑ | 73.95 \| 74.47 **76.83 \| 76.35** |
| | **MMAU-Pro-Music** | Gemini-2.5 Flash Music Flamingo | ACC ↑ | 64.90 **65.60** |
| | **MuChoMusic** | Qwen3-O Music Flamingo | ACC ↑ | 52.10 **74.58** |
| | **MMAR (Music)** | Qwen2.5-O Music Flamingo | ACC ↑ | 46.12 **48.66** |
| | **Music Instruct** | Audio Flamingo 3 Music Flamingo | GPT5 ↑ | 92.7 **97.1** |
| | **Music AVQA** | Audio Flamingo 3 Music Flamingo | ACC ↑ | **76.7** 73.6 |
| | **SongCaps (Ours)** *Human \| GPT5-Coverage \| GPT5-Correctness* | Audio Flamingo 3 Music Flamingo | Score ↑ | 6.5 \| 6.7 \| 6.2 **8.3 \| 8.8 \| 8.0** |
| **Music Information Retrieval** | **NSynth** *Source \| Instrument* | Audio Flamingo 3 Music Flamingo | ACC ↑ | 65.5 \| 78.9 **75.89 \| 80.76** |
| | **GTZAN** *Genre* | Pengi Music Flamingo | ACC ↑ | 80.00 **84.45** |
| | **Medley-Solos-DB** *Instrument* | Audio Flamingo 2 Music Flamingo | ACC ↑ | 85.80 **90.86** |
| | **MusicCaps** | Qwen3-O Music Flamingo | GPT5 ↑ | 7.2 **8.8** |
| **Lyrics Transcription** | **Opencpop** *Chinese* | GPT-4o \| Qwen2.5-O Music Flamingo | WER ↓ | 53.7 \| 55.7 **12.9** |
| | **MUSDB18** *English* | GPT-4o \| Qwen2.5-O Music Flamingo | WER ↓ | 32.7 \| 68.7 **19.6** |

*acknowledge the existence of numerous other MIR baselines and benchmarks, as MIR encompasses a broad range of tasks. For the scope of this paper, however, we restrict our comparisons to LALMs and the benchmarks most commonly used in the LALM literature. We encourage the community to further expand evaluations to a wider set of MIR baselines in future work.*

**Music Understanding and Reasoning Evaluation.** Table 1 shows that Music Flamingo consistently sets the bar across music QA, reasoning, MIR, and lyrics transcription benchmarks. On MMAU-Music, it reaches a competitive 76.83 accuracy, surpassing both closed and open-source models. The gap widens on the tougher MMAU-Pro-Music and MuChoMusic benchmarks, where Music Flamingo scores 65.6 and 74.58, respectively, a clear evidence of its robustness on complex datasets. Without reinforcement learning fine-tuning with thinking traces, performance drops to 63.9 and 69.5 respectively, highlighting the value of step-by-step reasoning and exploration. In MIR tasks, Music Flamingo continues to dominate: on NSynth, it achieves 80.76% accuracy in instrument recognition, and on Medley Solos DB, it reaches 90.86% for fine-grained instrument classification. It also delivers a significantly lower WER on Chinese and English lyrics transcription than both open and closed-source LALMs. These results establish Music Flamingo as the most capable model to date for detailed music understanding and reasoning.

On our proposed SongCaps benchmark, designed to evaluate music captioning, human raters scored the model outputs on a scale of 1-10. Music Flamingo achieves a high rating of 8.3 outperforming Audio Flamingo 3. Furthermore, we evaluate the captions using LLM-as-a-judge measuring both *correctness* and *coverage* of the caption. Music Flamingo achieves 8.0 for correctness and 8.8 for coverage, outperforming Audio Flamingo 3. These results highlight that Music Flamingo not only excels on structured QA and recognition tasks but also produces richer, more faithful natural language descriptions of music.

**Qualitative Evaluation.** We perform a thorough qualitative evaluation of Music Flamingo's outputs, assessed by trained music experts, in comparison with state-of-the-art LALMs in this domain. Due to

space constraints, we refer the readers to Appendix E for analysis on songs of varying genres and popularity, and additionally analysis of songs from different cultures in Appendix F.

## 5 CONCLUSION, LIMITATIONS AND FUTURE WORK

We introduced Music Flamingo, a large audio–language model designed to advance music understanding. By curating MF-Skills and MF-Think, we scale beyond short, instrumental clips to full-length, multi-cultural songs with layered annotations, and incorporate chain-of-thought reasoning for richer music analysis. Through improved pretraining, fine-tuning, and post-training with reinforcement learning, Music Flamingo achieves SOTA results across diverse music understanding and reasoning benchmarks. Beyond empirical gains, it demonstrates how models can move from surface-level recognition toward layered, human-like perception of songs.

Music Flamingo still has a few limitations, including: (i) limited understanding of underrepresented or skewed cultural traditions, highlighting the need to expand training data across more diverse global music; (ii) gaps in specialized tasks, such as fine-grained piano technique recognition and other instrument-specific skills; and (iii) the need to broaden coverage across additional musical skills to achieve more comprehensive understanding.

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

APPENDIX

## A   ETHICS STATEMENT

This work studies audio, music, and singing-voice understanding across culturally diverse material. Our experiments rely on publicly available datasets and/or content licensed for research use. We do not release copyrighted audio, stems, or lyrics; any examples used for qualitative illustration are either (i) already distributed by the originating dataset under a research-permissive license, or (ii) replaced by non-copyrightable descriptors (e.g., metadata, short transcriptions for analysis) when licenses are restrictive. No personally identifying information is collected, and no human-subjects experiments were conducted; institutional review board (IRB) approval was therefore not required.

**Cultural representation and bias.** Music corpora are uneven across regions, languages, and genres. Such imbalance can yield biased estimates or degrade performance on underrepresented traditions. We mitigate this by (a) documenting dataset composition and selection criteria, (b) emphasizing vocal and multicultural material in evaluation, and (c) reporting known limitations. We encourage downstream users to avoid normative claims about "quality" across cultures and to treat our benchmarks as descriptive rather than prescriptive.

**Copyright and content ownership.** Models trained on musical recordings risk reproducing protected content. We do not deploy or evaluate generative audio synthesis; our outputs are textual (QA, captions, reasoning traces). We avoid releasing any asset that could enable reconstruction of substantial portions of copyrighted works and provide guidance for filtering long verbatim lyric reproduction in evaluation outputs.

**Privacy and safety.** Singing voices may implicitly encode sensitive traits. We use only public research datasets and focus on musical attributes (rhythm, harmony, timbre, structure) rather than identifying individuals. Potential misuse includes intrusive listener profiling or surveillance via audio analysis; to discourage these, we release only research artifacts (documentation, evaluation protocols, non-identifying metadata) and clearly scope permitted use in licenses when possible.

## B   REPRODUCIBILITY STATEMENT

We provide all details needed to reproduce our results within the paper and appendix: dataset sources and splits; audio preprocessing (sampling rates, normalization, chunking/segment lengths); model architectures and parameter counts; training schedules (optimizers, learning-rate policies, batch sizes, gradient clipping), GRPO/post-training settings and reward definitions; inference settings (temperature, decoding constraints); and exact evaluation protocols and metrics for every benchmark. We report hardware used where applicable, and mean±std for repeated trials. We will release code, checkpoints and data upon acceptance.

## C    MUSIC FLAMINGO TRAINING DATASETS

Table 2 summarizes all datasets used to train Music Flamingo, including total hours, number of audio-QA pairs, and the number of epochs (passes over the dataset) used at each training stage. Similar to (Ghosh et al., 2025; Goel et al., 2025), we convert all foundational datasets (captioning, classification, etc.) into QA formats, using the same set of prompts for each task mentioned in (Ghosh et al., 2025; Goel et al., 2025).

Table 2: List of fine pre-training and fine-tuning datasets together with their training composition.

| Dataset | Hours | Num. Pairs | AF3-St. 3 | MF-SFT | MF-Warmup | MF-GRPO |
|---|---|---|---|---|---|---|
| AF3-training mix (Goel et al., 2025) | - | 30M | 1.0 | - | - | - |
| MF-Skills (Ours) | - | 3M | - | 2.0 | - | 1.0 |
| MF-Think (Ours) | - | 176k | - | - | 1.0 | 1.0 |
| MusicBench (Melechovsky et al., 2023) | 115.5 hrs | 686k | 1.0 | 1.0 | - | - |
| Mu-LLAMA (Liu et al., 2024a) | 62.9 hrs | 70k | 1.0 | 1.0 | - | - |
| MusicAVQA$_{audio-only}$ (Li et al., 2022) | 77.1 hrs | 5.7K | 1.0 | 1.0 | - | - |
| MusicQA (Ouyang et al., 2025) | 62.9 hrs | 70K | 1.0 | 1.0 | - | - |
| LP-MusicCaps$_{MSD}$ (Doh et al., 2023) | 5805.7 hrs | 1331.8K | 1.0 | 1.0 | - | - |
| LP-MusicCaps$_{MTT}$ (Doh et al., 2023) | 126.4 hrs | 46.9K | 1.0 | 1.0 | - | - |
| LP-MusicCaps$_{MC}$ (Doh et al., 2023) | 7.4 hrs | 7.9K | 1.0 | 1.0 | - | - |
| MusicCaps (Agostinelli et al., 2023b) | 7.4 hrs | 2.6K | 1.0 | 1.0 | - | - |
| NSynth (Engel et al., 2017) | 321.3 hrs | 289.2K | 1.0 | 1.0 | - | - |
| MusDB-HQ (Rafii et al., 2017) | 29.1 hrs | 10.2K | 1.0 | 1.0 | - | - |
| FMA (Defferrard et al., 2016) | 860.7 hrs | 104.2K | 1.0 | 1.0 | - | - |
| Music4All Captions (ours) | 910.5 hrs | 109k | 1.0 | 1.0 | - | - |
| Music4All QA (ours) | 1505.7 hrs | 180k | 1.0 | 1.0 | - | - |
| MSD Captions (ours) | 15449.9 hrs | 1.4M | 1.0 | 1.0 | - | - |
| MSD QA (ours) | 20906.2 hrs | 935k | 1.0 | 1.0 | - | - |
| CHIME (Foster et al., 2015) | 342 hrs | 30k | 1.0 | - | - | - |
| ALI Meeting (Yu et al., 2022) | 118.75 hrs | 387k | 1.0 | - | - | - |
| EMILIA (He et al., 2024) | 5000 hours | 1.7M | 1.0 | - | - | - |
| MUST (Qin et al., 2025) | 500 hrs | 245k | 1.0 | - | - | - |
| CoVoST (Wang et al., 2020) | 2880 hrs | 5M | 1.0 | - | - | - |
| Multi-talker Switchboard (Godfrey et al., 1992) | 109.9 hrs | 76.6K | 1.0 | - | - | - |

## D    MUSIC FLAMINGO TRAINING DETAILS

In this section, we present the training settings of our model across all stages, each with specific configurations. Details are in Table 3.

| Settings | AF3-SFT | MF-SFT | MF-WarmUp | MF-GRPO |
|---|---|---|---|---|
| global batch size | 128 | 128 | 128 | 64 |
| learning rate | 1.5e-5 | 1.5e-5 | 1e-5 | 1e-6 |
| learning schedule | | | Cosine decay | |
| warm up ratio | | | 0.03 | |
| weight decay | | | 0.0 | |
| epoch | 1 | 1 | 1 | 1 |
| bf16 | ✓ | ✓ | ✓ | ✓ |
| grad accumulate | | | 8 | |
| FSDP – full shard | ✓ | ✓ | ✓ | ✓ |
| GPUs | | | 128×A100 | |

Table 3: Training settings across stages.

## E    USER STUDY ON MUSIC FLAMINGO

We undergo a user study with trained music experts comparing Music Flamingo to an open-source LALM–*Qwen3 Omni*, and two closed-source LALMs–(*GPT-4o* and *Gemini 2.5 Pro*) qualitatively. To achieve this, we selected a subset of 8 songs: 4 songs in English and 4 songs in Brazilian Portuguese. From these songs, half of them are by extremely popular artists in Western music, and other half from

| Aspect | Music Flamingo | Qwen3-Omni | GPT4o-Audio | Gemini-2.5 Pro |
|---|---|---|---|---|
| **General technical characteristics (tempo, key, time signature)** | Consistently outputs tempo (bpm) and key; sometimes time signatures (3 songs correctly 4/4). Mistakes usually in relative major/minor. | Rarely detailed; often omits tempo/key/time signature. | Sometimes outputs tempo (bpm usually in ballpark). Less consistent with key. | Sometimes outputs tempo (closest bpm for *NO I DON'T*). Some key mistakes. Time signatures generally omitted. |
| **Genre classification** | Reasonably good, but misclassified *Maniac* (electro-funk vs. dance-pop). Struggled with Brazilian music genres (occasional mismatches). | Superficial; often wrong with Brazilian genres. | Some correct, but not consistent. | Best at identifying genres overall, but hallucinated (*ABBA* as ska cover; *Wild Things* as dream pop with drums). |
| **Emotional content & lyrics** | Good understanding; captures emotional context. Lacks cultural/historical nuance. | Similar to Music Flamingo; misses deeper cultural context. | More superficial understanding than others. | Good understanding; one small lyric mistake. Also lacks cultural/historical nuance. |
| **Complex technical characteristics (chord progressions, structure, production)** | Attempts deeper detail, but sometimes inaccurate with chords/voicings/structure. Sometimes hallucinates genre-related elements. | Least detailed; only superficial composition/arrangement notes. | Gives more detail when recognizing famous songs; otherwise shallow. | More detailed for famous songs; hallucinations (e.g., nonexistent drums, overstated synths). Genre misclassifications cascade into wrong technical details. |
| **General observations** | Strong in consistent technical reporting, but accuracy varies on deeper features. | Shallowest outputs. | Relies on recognition of famous songs; may pull from text knowledge instead of audio. | Similar to GPT4o-Audio; detailed when recognizing famous songs. Most prone to hallucinations tied to genre assumptions. |

Table 4: Comparison of Music Flamingo, Qwen3-Omni, GPT4o-Audio, and Gemini 2.5 Pro across different evaluation aspects.

less known artists. The following songs were used: 1) *ABBA - Money Money Money*, 2) *Michael Sembello - Maniac (From Flashdance)*, 3) *Chandler Leighton - NO I DON T*, 4) *Lø Spirit - Wild Things*, 5) *Antônio Carlos Jobim - Águas De Março*, 6) *Michel Teló - Ai Se Eu Te Pego*, 7) *Paulinho da Viola - Apoteose Ao Samba* and 8) *Ave Sangria - Seu Waldir*.

Table 4 shows a summary of the detailed analysis comparing different musical aspects and features across models. Among the four models, Music Flamingo performs the best overall while some limitations in accurately identifying deeper context remain.

## F    COMPARATIVE ANALYSIS ACROSS SONGS FROM DIFFERENT CULTURES

Furthermore, we compare the strengths of Music Flamingo on five commercially released songs spanning cultures, languages and styles: Niuver *Enamorados* (Spanish, Latin ballad) (Figure 5), Annika Wells *Jim & Pam* (English, indie/acoustic pop)(Figure 6), Louane *La fille* (French, piano-led pop)(Figure 7), Michel Telo *Ai Se Eu Te Pego* (Portuguese, Brazilian sertanejo)(Figure 8), and Zemlyane *Trava u doma* (Russian, Soviet rock)(Figure 9). Below we present a detailed summary of this comparison.

**General technical characteristics (tempo, time signature, key).** MF consistently produced numeric tempos and keys that matched canonical analyses or widely observed half/double-time readings, and it explicitly handled relative-minor vs. metadata-major ambiguities (e.g., *Enamorados*: metadata in C major while harmony centers on A minor). GPT-4o and Gemini often described tempo qualitatively or gave numeric ranges but omitted keys; when numeric BPMs were provided, both models occasionally drifted toward club-tempo values that better reflect remixes than the canonical singles (e.g., *Ai Se Eu Te Pego*: 128–140 BPM claimed vs. ∼96 BPM on the hit version). Qwen3 frequently omitted numerics altogether. Time signature was rarely stated by any model; where implied, 4/4 matched all five tracks.

*Illustrative cases.* (i) *Jim & Pam*: MF reported the double-time ∼158 BPM and the correct key (D), aligning with a felt pulse at ∼79 BPM; GPT-4o gave the correct qualitative tempo band but no key; Gemini mis-estimated to 120–125 BPM and mis-keyed E. (ii) *La fille*: MF aligned with ∼128 BPM in C; Gemini and GPT-4o underestimated (90–100 BPM) and/or mis-keyed. (iii) *Trava u doma*: MF matched A minor and ∼130 BPM; GPT-4o underestimated to ∼100–110 BPM.

**Genre.** All models could identify the broad style family. *Gemini* held a *narrow* edge on matching canonical catalog labels and regional taxonomy (e.g., *Ai Se Eu Te Pego*: *sertanejo universitário* with dance-pop trappings). MF was directionally correct across the set and, in two cases, selected closely related tags when salient timbres or arrangement scale were misleading (*Ai Se Eu Te Pego*: forró inferred from accordion timbre; *Trava u doma*: prog-leaning language for a synth/space-colored Soviet pop/rock record). Importantly, these adjacent picks did not derail MF's downstream harmony/structure reasoning and are straightforward to normalize to catalog labels. *GPT-4o* typically described the stylistic *feel* accurately (e.g., dance-pop with Brazilian flair; piano-led ballad) but often stopped short of naming the canonical label. *Qwen3* alternated between sensible tags (French indie/pop ballad) and broad era styles ("80s arena/soft rock"), and twice misframed vocal songs as instrumental (see below), which contaminated the subsequent genre claim.

*Takeaway.* Canonical label accuracy: **Gemini** ≳ **MF** ≈ **GPT-4o** > **Qwen3**. Gemini's advantage is mostly in verbatim catalog taxonomy; MF's labels are correct at the family level and, when adjacent, remain musicologically consistent with its superior harmonic/structural analysis.

**Emotional content and lyrics.** MF, Gemini, and GPT-4o gave coherent, text-grounded readings of mood and themes across all songs (e.g., *Enamorados*: memory, time, and fading love; *La fille*: identity and self-doubt; *Trava u doma*: homesick cosmonaut narrative). Where lyrics were quoted or paraphrased, all three remained faithful to content and tone. Qwen3 produced reasonable affect reads when it acknowledged lyrics, but twice declared a vocal track "instrumental" (*Enamorados*, *Trava u doma*), leading to incorrect conclusions about narrative and emotion.

*Observation.* When models inferred emotion strictly from sonics without anchoring in lyric text, nuance decreased and culture-specific references were missed (e.g., *Trava u doma* as an iconic space-age anthem; *Ai Se Eu Te Pego* as a global sertanejo earworm driven by chant-like hooks).

**Complex technical characteristics (chord progressions/voicings, song structure, production).** MF generally provided the deepest harmonic/structural content (naming cadential behavior, relative-minor centers, verse/chorus dynamics), and its structural reads were consistently plausible across all five songs. Its main failure mode was *over-specification*: occasionally asserting colorful altered/extended chords or percussion layers not supported by public charts or by the stems one would expect (*Jim & Pam*: introduced drum-machine and synth-bass in an otherwise hand-clap/snaps, acoustic texture; *La fille*: added brushed kit and altered dominants to a piano-centric, drum-light mix). GPT-4o's arrangements and sectioning were reliably correct (intro/verse/chorus/bridge placement,

dynamic swells), with conservative but accurate production notes; it rarely named specific harmonic content, which limited precision but avoided hallucination. Gemini's arrangement commentary was serviceable and sometimes quite apt on famous tracks (accordion/synth hook in *Ai Se Eu Te Pego*), but often remained generic and light on concrete harmony. Qwen3's technical layer was the sparsest and suffered when the top-level premise was wrong ("instrumental"), cascading into inapplicable structure/production claims.

*Failure-mode coupling.* We repeatedly observed that **genre misclassification leads to production hallucinations**. For instance, mapping *Ai Se Eu Te Pego* to *forró* primed mentions of forró-typical percussion, and reading *La fille* as an indie/pop ballad with a "soft electronic beat" invited non-existent drum programming. Conversely, when models named the *canonical* genre, instrumentation and mix notes tended to be accurate (Gemini on *Ai Se Eu Te Pego*; MF and GPT-4o on the piano+voice core of *La fille*).

**Model-specific observations (holistic).**

- **Music Flamingo.** Strongest on measurable facts (tempo/key) and the only model to consistently reconcile metadata major vs. relative-minor centers (*Enamorados*: C vs. A minor). Best overall at section-level reasoning and harmonic intent, but occasionally over-decorates with color chords or speculative percussion.

- **Gemini 2.5 Pro.** Best at canonical genre taxonomy and culturally grounded framing (e.g., *sertanejo universitário*). Tends to under-specify harmony and sometimes over-estimate tempo; instrumentation is usually correct when genre is correct.

- **GPT-4o.** Most dependable for arrangement, dynamics, and production prose; excellent at emotion/lyric grounding. Hesitant on numerics (tempo/key) and sometimes avoids naming the canonical genre even when its prose implies it.

- **Qwen3-Omni.** Capable of coherent mood and arrangement reads when anchored, but inconsistent. The two "instrumental" misclassifications (on clearly vocal songs) demonstrate a brittle failure that propagates into wrong genre/production claims.

Overall, the pattern is clear: *precision on the measurable (tempo/key) + correct canonical genre naming* is the foundation for faithful structure/production descriptions and culturally aware, lyric-grounded emotion reads. Among the models studied, **Music Flamingo** delivers the most reliable technical grounding and music-theoretic reasoning; **Gemini** adds the most accurate taxonomy; and **GPT-4o** supplies consistently correct structural/production narratives.

---

**Caption Evaluation Prompt: Niuver - Enamorados (Q7gVr-VMfKo)**

Describe this track in full detail — tell me the genre, tempo, and key, then dive into the instruments, production style, and overall mood it creates.

**Output**

This track is a Latin American folk ballad that blends traditional acoustic storytelling with a contemporary singer-songwriter aesthetic. It moves at a moderate 107 BPM and is rooted in A minor, giving the piece its wistful, minor-toned character.

Instrumentation & production: The arrangement is built around a warm, resonant nylon-string acoustic guitar that carries the harmonic and melodic material, complemented by a subtle acoustic bass that grounds the low end. Light percussion—soft brushes and occasional shakers—adds a gentle pulse without breaking the intimacy. The production is deliberately organic and natural-sounding: instruments are recorded close-miked, with minimal processing, and the mix places the guitar and vocals front-and-center in a balanced stereo field. The overall sound is clean, with a warm analog-type EQ curve and modest compression that preserves the dynamic nuance of each phrase.

Vocal characteristics: A female mezzo-soprano delivers the lyrics in Spanish with a clear, warm timbre. Her delivery is intimate and expressive, featuring a slight vibrato and a storytelling style that emphasizes the emotional weight of each line. The vocal track is treated with only light reverb and subtle compression, allowing the natural dynamics and nuance of her performance to shine through.

Lyrical themes: The lyrics contemplate the passage of time, lost love, and the bittersweet nature of memory. Repetitive, hook-like lines such as "Enamorados, enamorados, cambia y cambia la ilusión, mi amor, con el pasado" underscore the central theme of love that has faded and the changing illusion of the past. Verses like "Cómo ha pasado ese tiempo… ¿De qué he pasado, mi alegro?" and "Perder el tiempo de un beso… Cuando se pierde en un sueño" deepen the sense of longing and reflection.

Song structure & dynamics: The composition follows a classic verse-chorus form. An intro of solo guitar establishes the tonal center before the first verse, which introduces the narrative. The chorus arrives with the repeated "Enamorados…" refrain, heightening emotional intensity while maintaining the same gentle dynamic level. A second verse expands the story, followed by another chorus that reinforces the central motif. The track concludes with a brief instrumental outro that mirrors the intro, allowing the piece to fade out on the lingering resonance of the guitar.

Theoretical grounding: Harmonic movement centers on the A-minor tonic, with frequent use of the dominant E7 (and its variations E7/G#) to create tension that resolves back to Am. The progression also incorporates modal mixture and secondary dominants—e.g., Bm7♭5 and F7—adding color and a subtle sense of yearning. The recurring Am → E7 → Am loop underpins the verses, while the chorus leans on a brighter shift to C-major-related chords (Cmaj7, Fmaj7) before returning to the minor tonic, providing a modest lift that mirrors the lyrical contrast between memory and present loss.

Overall mood & context: The song exudes a melancholic, nostalgic atmosphere, evoking the quiet introspection of a late-evening acoustic session in a small Latin American town. Its blend of folk instrumentation, heartfelt Spanish vocals, and reflective lyricism situates it within the contemporary revival of traditional Latin American ballads, appealing to listeners who appreciate authentic storytelling set against a gently flowing acoustic backdrop.

---

Figure 5: Caption generated by Music Flamingo on a modern Spanish song.

**Caption Evaluation Prompt: Annika Wells - Jim & Pam (7nt-d3bAZqM)**

Write a rich caption that blends the technical details (genre, BPM, key, chords, mix) with how the song feels emotionally and dynamically as it unfolds.

**Output**

This track is an Indie Pop song with a strong Bedroom Pop aesthetic, blending lo-fi, intimate production with a bright, melodic pop sensibility. It moves at a brisk 157.9 BPM and is rooted in D major, giving the piece a sunny, major-key feel.

Instrumentation & production: The arrangement is built around a clean, arpeggiated electric guitar that carries the harmonic texture, a simple drum-machine pattern that provides a steady, understated pulse, and a warm synth bass that underpins the low end. The mix is deliberately narrow-stereo and lo-fi, emphasizing a cozy, bedroom-studio vibe; reverb is kept subtle, and the overall mastering retains a gentle, uncompressed character that lets the vocals sit front-and-center.

Vocal characteristics: A female lead delivers the lyrics with a soft, clear, slightly breathy timbre. Her delivery is conversational and intimate, using minimal processing—just a touch of reverb—to preserve the natural, vulnerable quality of her voice.

Lyrical themes: The lyrics celebrate a deep, affectionate friendship that borders on romance, using pop-culture references ("you're my Jim, I'm your Pam") to frame the bond. The chorus repeats the hook:

> "Cause you're my Jim, and I'm your Pam, I fell in love with my best friend.
> No one's ever loved me like you do.
> Cause no one's Jim and Pam like me and you."

Verses add playful, everyday details ("I'll leave you sleeping on my shoulder… I'll switch the teapot for the iPod…") that reinforce the theme of unwavering support and shared intimacy.

Song structure & dynamics: The composition follows a clear verse-chorus layout with an introductory guitar arpeggio, two verses, a recurring chorus, and a brief bridge that introduces a slight harmonic shift before returning to the final chorus. Dynamics stay consistently soft; the track relies on melodic and lyrical variation rather than loud-soft contrasts to maintain interest.

Harmonic analysis: The chord progression cycles through D-major-related colors—Dmaj7, Gmaj7, Bm, and occasional chromatic moves such as C#7/E# and F#7—creating a bright, uplifting backdrop while the occasional minor chords (Bm, Em) add a gentle, reflective shade. This mixture of major-tonic stability and subtle minor inflections supports the song's warm, affectionate mood.

Overall mood & context: The song exudes a cozy, affectionate atmosphere, perfect for late-night listening or a relaxed gathering. Its lo-fi bedroom-pop production places it squarely within the 2020s indie scene, where intimate storytelling and minimalist aesthetics are celebrated.

Figure 6: Caption generated by Music Flamingo on a less known American song.

---

**Caption Evaluation Prompt: Louane - La fille (sqK71EWTs4I)**

Break the track down like a critic — list its tempo, key, and chordal motion, then explain the textures, dynamics, and emotional impact of the performance.

**Output**

This track is a melancholic French Pop / Indie Pop piece that blends intimate singer-songwriter sensibilities with subtle electronic-ambient textures. It moves at a moderate 130.43 BPM and is rooted in C major, though the harmonic language leans heavily toward minor-toned colors, reinforcing its introspective mood.

Instrumentation & production: The arrangement is built around a warm, resonant piano that carries the main melodic material, complemented by airy synth pads that add depth without overwhelming the acoustic core. A soft, brushed-style drum kit provides a gentle pulse, while a smooth electric bass underpins the harmony with a rounded, low-mid presence. The mix is clean and spacious, employing a wide stereo field that places the piano and synths slightly off-center, the bass anchored in the center, and the vocals perched atop a subtle reverb-and-delay tail. Mastering emphasizes clarity and dynamic range, allowing the quiet verses to breathe and the choruses to lift without compression-induced harshness.

Vocal characteristics: The lead is a French-speaking female mezzo-soprano. Her timbre is clear, slightly breathy, and highly emotive, delivering the lyrics with a vulnerable, introspective tone. The vocal track is treated with modest reverb and delay, creating an ethereal ambience that matches the song's reflective atmosphere.

Lyrical themes: The lyrics explore self-identity, perception, and the tension between external expectations and inner reality. Repetitive, hook-like lines such as "*C'est qui la fille*" ("Who is the girl") and "*C'est p't-être un peu loin de la vérité*" ("It might be a bit far from the truth") underscore a sense of searching and uncertainty. The chorus repeatedly asks the listener to consider what they see versus what they believe, reinforcing the theme of personal doubt.

Song structure & dynamics:
- Intro: Sparse piano arpeggios with ambient pads set a contemplative tone.
- Verse 1: Minimal drum brush work and bass, allowing the vocal narrative to unfold.
- Pre-chorus: Slightly richer harmonic texture (introducing Fmaj7 and G chords) builds tension.
- Chorus: Full-band entry with layered synths, a more pronounced drum groove, and the repeated "*C'est qui la fille*" hook, delivering an emotional lift.
- Verse 2 / Bridge: Returns to a stripped-down arrangement, then introduces a brief bridge featuring a shift to the relative minor (Am) and a subtle harmonic surprise (Bm7b5/A) that deepens the sense of yearning.
- Final chorus & outro: Re-expands the arrangement, adding vocal harmonies and a gentle fade-out of the piano and pads, leaving a lingering sense of unresolved introspection.

Theoretical grounding: Although the piece is in C major, the chord progression frequently emphasizes the vi (Am) and ii (Dm) chords, creating a minor-leaning tonal center. The recurring movement from C → Am → Fmaj7 → G (I-vi-IV-V) provides a classic pop-song cycle that balances resolution with melancholy. The occasional use of Bm7b5/A and G7 introduces a brief modal mixture, heightening the emotional tension before resolving back to the tonic C, mirroring the lyrical quest for self-definition.

Overall mood & context: The track exudes a wistful, vulnerable aesthetic that situates it within the contemporary French indie-pop scene, where lyrical introspection and minimalist production are prized. Its blend of piano-driven pop songwriting with ambient synth layers reflects a broader 2020s trend of merging organic instrumentation with subtle electronic textures to convey personal, confessional narratives.

---

Figure 7: Caption generated by Music Flamingo on a modern French song.

---

**Caption Evaluation Prompt: Michel Teló - Ai Se Eu Te Pego (hcm55lU9knw)**

Write a rich caption that blends the technical details (genre, BPM, key, chords, mix) with how the song feels emotionally and dynamically as it unfolds.

**Output**

This track is a high-energy Brazilian Forró piece, rooted in the traditional dance-hall style and flavored with a modern, live-performance vibe. It sits firmly in the Forró sub-genre, blending the genre's classic accordion-driven rhythm with a driving electric-bass groove and punchy drum kit.

Tempo & Key – The song moves at ≈ 93.75 BPM and is centered in B major . (No time-signature information was supplied.)

Instrumentation & Production – The arrangement is built around a bright, melodic accordion that carries the main melodic hooks, supported by a driving electric bass that locks in with a punchy drum kit (kick, snare, hi-hats) to keep the danceable pulse. The production is deliberately raw and live-sounding , with a wide stereo field that places the accordion slightly left, the bass and drums centered, and crowd ambience spread across the mix, giving the impression of a festive night-out performance. Minimal processing is applied; the instruments are captured with natural room reverb and modest compression to retain dynamics while keeping the overall level high.

Vocal Characteristics – The lead vocalist is a male tenor with a clear, slightly raw timbre. His delivery is energetic and conversational, often engaging in call-and-response with backing singers. The vocals are sung in Brazilian Portuguese , treated with light reverb and compression to sit prominently in the mix without sounding overly polished.

Lyrical Themes – The lyrics revolve around a playful, flirtatious encounter at a Saturday night party. Repetitive, chant-like lines such as "Nossa, nossa, assim você me mata" and "Ai, se eu te pego" serve as the song's hook, reinforcing the teasing, romantic pursuit. Verses describe the scene ("Sábado na balada… A galera começou a dançar…") and the narrator's bold approach to the "menina mais linda," creating a light-hearted, party-centric narrative.

Song Structure & Dynamics – The form follows a verse-chorus layout with instrumental breaks that showcase the accordion and bass. An intro establishes the rhythmic groove, followed by verses that build tension through vocal phrasing. The choruses explode with the repeated hook, while mid-song instrumental sections provide a brief release before returning to the vocal refrain. A final chorus and a short outro round out the track, maintaining the high-energy momentum to the end.

Theoretical Insight – Harmonic movement largely orbits the B-major tonic , with frequent use of the dominant F♯ and sub-dominant E chords, creating a bright, forward-moving feel. Occasional minor-quality chords (e.g., Abm , Ebm ) add a fleeting, slightly wistful color before resolving back to the major tonality, reinforcing the song's upbeat yet flirtatious mood.

Overall Mood & Context – The overall aesthetic is festive, lively, and irresistibly danceable , capturing the spirit of a Brazilian Saturday night in a bar or festa. Its raw, live-room production and crowd ambience place it within the contemporary revival of traditional Forró, appealing both to fans of classic regional music and to younger audiences seeking an energetic party anthem.

---

Figure 8: Caption generated by Music Flamingo on a modern Brazilian song.



**Caption Evaluation Prompt: Земляне - Трава у дома (C4Pajis7OOU)**

```
Describe this song from both a technical and artistic lens: mention tempo, harmony, and
instrumentation, but also mood, lyrical themes, and structure.
```

**Output**

```
This track is a Russian Rock piece that leans heavily into Progressive Rock, blending
classic-rock guitar-driven energy with the expansive, shifting arrangements typical of prog.

Tempo & key - The song moves at a steady 130.43 BPM and is rooted in A minor.

Instrumentation & production - The arrangement is built around a tight acoustic drum kit and a
solid electric bass that lock into a driving groove. Multiple electric guitars dominate the
texture: clean arpeggiated lines, overdriven rhythm chords, and soaring lead solos that weave
through the harmonic changes. Atmospheric synth pads add depth and a slightly futuristic sheen.
The mix is polished and high-fidelity, with a wide stereo field that places the guitars and
synths on opposite sides while the vocals sit centrally, all treated with tasteful reverb and
delay for spaciousness.

Vocal characteristics - A male baritone delivers the Russian lyrics with a clear, slightly
melancholic timbre. His delivery is melodic and narrative, emphasizing the song's reflective
mood. The voice is processed with reverb and delay, giving it a spacious, almost ethereal
quality.

Lyrical themes - The lyrics explore longing for home and the contrast between the harshness of
space and the comforting memory of Earth. The recurring chorus line—*"И снится нам не рокот
космодрома, не та ледяная синева, а снится нам трава, трава у дома, зелёная, зелёная трава"*—
highlights the central image of a green meadow versus the cold, metallic sounds of a rocket
launch. Verses describe the view of Earth through a porthole, the weightlessness of orbit, and
the emotional distance from family, reinforcing the theme of yearning for familiar, natural
beauty.

Song structure & dynamics - The composition follows a clear verse-chorus layout with an
introductory guitar motif, followed by verses that build tension through layered guitars and
synths. Each chorus erupts with the full band, reinforcing the melodic hook. Instrumental
breaks feature extended guitar solos that develop the thematic material, while a bridge
introduces a brief harmonic shift before returning to the final, anthemic chorus and a decisive
outro. Dynamic intensity rises from the restrained verses to the full-throttle choruses and
solos, then gently recedes at the close.

Theoretical grounding - Harmonic movement centers on A minor, with frequent use of the dominant
E and sub-dominant D chords (e.g., Am - E - D) that create a sense of tension and release. The
progression often passes through F and G major chords, providing brief modal lifts before
returning to the minor tonic, a common prog-rock technique that underscores the song's
searching, melancholic mood.

Overall mood & context - The track exudes a reflective, slightly melancholic atmosphere,
marrying the earnest storytelling of Russian rock with the sophisticated arrangements of
progressive rock. Its polished production and expansive instrumentation place it within the
modern Russian prog-rock tradition, echoing the 2000s-era revival of concept-driven, emotionally
charged rock music.
```



Figure 9: Caption generated by Music Flamingo on a well known Russian song.

## G    LINEAR PROBING EXPERIMENTS WITH AUDIO ENCODERS

In order to better understand the role of the audio encoder in music tasks, we performed linear probing experiments with three audio encoders: the audio encoder from Qwen2Audio (Chu et al., 2024) and Audio Flamingo 3 (Goel et al., 2025) – the encoder on which Music Flamingo is based, which are both based on the Whisper architecture, and the MERT encoder (Yizhi et al., 2023), which has an architecture with a bias towards understanding music due to its use of a constant-Q transform representation in the input. We chose two tasks from the MARBLE benchmark (Yuan et al., 2023): the key classification task using the GS dataset (Knees et al., 2015), and the genre classification task using the GTZAN dataset (Tzanetakis & Cook, 2002). All linear probing models were trained using the average of all the frames from the audio representation in question (Qwen2Whisper, AFWhisper, or MERT) with a single linear layer. Models were trained until no improvement in validation set accuracy was observed after 5 epochs, and validation set accuracy was used as well to choose the best checkpoint. Results can be found in Table 5.

Table 5: Performance comparison on key and genre classification with different audio encoders.

| Model | GS (key cls.)↑ | GTZAN (genre cls.)↑ |
|---|---|---|
| Qwen2Audio | 34.10 | 89.99 |
| AFWhisper | 40.56 | **91.37** |
| MERT | **56.12** | 78.96 |

We observe that both models based on Whisper, which were trained specifically for captioning, have high accuracy for genre classification but comparatively low accuracy for the key classification task. We hypothesize this is because when using captioning targets, the likelihood of having the genre for a given song mentioned in the caption is much higher than having the key the song is in. MERT, on the other hand, has lower accuracy for the higher level task but higher accuracy for the lower level task. MERT is trained to generate a representation that can be used to reconstruct the audio through a VAE "acoustic teacher" model, and also to match the coefficients of a CQT transform (the "music model"), therefore being a more lower level representation than text-based captions.

Given the correlation between this gap in accuracy between lower and higher level tasks and the general observations for the outputs of Music Flamingo, in the future we intend to try an audio encoder that preserves more of the lower level information in audio to enable future versions of the model to perform better in understanding those features from music signals.

### G.1 EXAMPLES OF MF-SKILLS AND MF-THINK

*MF-Skills* contains long, multi-aspect targets (captions and QA) *without* reasoning traces. Figure 10, Figure 11, Figure 12, Figure 13, Figure 14 shows examples of captions in the MF-Skills dataset. *MF-Think* augments a challenging subset with compact chain-of-thought traces (`<think>...</think>`) and final answers (`<answer>...</answer>`). Figure 15, Figure 16, Figure 17, Figure 18, Figure 19 and Figure 20 shows examples of captions and QA pairs in the MF-Think dataset.

---

**Caption Evaluation Prompt: DRIMS - Frío (M5pAw4l6HYA)**

Give me a full musical picture: what's the style, BPM, and harmonic palette, and how do the vocals, instrumentation, and production shape its mood?

**Output**

This track is a Latin Alternative Rock ballad that fuses the melodic sensibility of Latin pop with the raw emotional edge of Emo Rock.

At ≈136 BPM (4/4 feel) and set in E major , the arrangement opens with a delicate acoustic-guitar intro that establishes a gentle, open-string texture before the full band erupts. Distorted electric guitars layer crunchy power-chords and bright, ringing voicings (Amaj7, A6, B7, F♯m7, C♯m7), while a soaring lead-guitar solo adds a melodic climax. The rhythm section—tight, driving drums with a punchy kick and crisp snare, and a warm, rounded bass—propels the song forward, and subtle stereo-wide reverbs on the guitars create a spacious, high-fidelity mix. Production is polished, with careful EQ separation that lets the acoustic intro sit intimate, then expands into a full-band "wall of sound" for the choruses and solo.

Vocals are delivered by a male tenor whose timbre is warm and resonant. He sings primarily in a strong chest voice, occasionally slipping into falsetto for emotional swells; reverb and delay are applied to accentuate the melancholy and longing. The vocal style is clean, melodic, and highly expressive, matching the lyrical content.

Lyrically the song dwells on a lingering, cold-filled memory of a past love. The chorus repeats the central motif:

- " Y hoy que hace tanto frío, no quiero flores de papel. "
- " Y hoy, mirando al vacío, recuerdo la primera vez que en frío te esperé. "

and later: " Y hoy, ahogándome en el frío, recuerdo esa primera vez que de ti me enamoré. "
The verses deepen the narrative ("Estoy un poco confundido… Yo y tú, vestida de amarillo…") while the outro reinforces the resolve " Nunca te olvidaré ." The repeated references to "frío" (cold) and "vacío" (void) underscore a theme of frozen longing and the inability to let go.

Structurally, the song follows a classic ballad form:
- Intro (acoustic guitar) → Verse 1 → Pre-chorus (building dynamics) → Chorus (full band, harmonic lift) → Verse 2 → Chorus → Interlude/solo (electric-guitar lead) → Bridge (reduced texture, vocal focus) → Final Chorus (maximal intensity) → Outro (soft vocal echo). Each section adds layers, moving from intimate to expansive, then tapering back to a gentle close.

The harmonic progression emphasizes the tonic E major while frequently moving to its subdominant (A) and dominant (B7) with extended colors (Amaj7, A6, F♯m7, C♯m7). These shifts create a wistful, searching mood in the verses that resolves into brighter, more anthemic passages in the choruses, reinforcing the emotional release of recalling a cherished but lost love.

Overall, the track exudes a bittersweet, melancholic aesthetic, situating itself within the early-2020s wave of Latin-American bands that blend rock's intensity with emo-style lyricism, offering both cultural resonance and universal heartbreak.

Figure 10: Example of a ground-truth caption on a Latin song in MF-Skills.

# H   PROMPTS FOR MF-SKILLS AND MF-THINK

We provide all prompting templates used across our datasets and QA types in Figures 21, 22, 23, 24, 25, 26.

**Caption Evaluation Prompt: Benjamin Burrell - Lay In Wait (part 3) (17dYKXJo5BI)**

Give me a full musical picture: what's the style, BPM, and harmonic palette, and how do the vocals, instrumentation, and production shape its mood?

**Output**

This track is a neo-classical, cinematic instrumental piece that blends contemporary classical piano writing with ambient synth pads and a lush orchestral palette. At an estimated 88 BPM (though the rhythm is free-flowing rubato with no fixed time signature), it is rooted in C♯ major, while the harmonic language leans heavily toward minor tonalities, featuring progressions such as C♯ m – F♯ m – B m and occasional chromatic shifts (e.g., F♯6, Bb/C) that create a searching, melancholic atmosphere.

Instrumentation is dominated by a resonant grand piano that carries the melodic narrative, supported by ethereal synthesizer pads that fill the stereo field, sweeping string sections that provide both harmonic depth and a sustained low-brass foundation. The production is high-fidelity and spacious, with a wide stereo image that places the piano centrally while the pads and strings envelop the listener, giving the piece a cinematic, immersive quality. Mixing emphasizes clarity and depth, allowing the delicate piano articulation to sit above the warm, reverberant orchestral backdrop.

There are no prominent vocals; the composition is entirely instrumental.

The lyrical (thematic) content is expressed through the music itself: a deep sense of longing, reflective melancholy, and serene beauty. The gradual build from a fragile piano intro into powerful orchestral swells, followed by a gentle, contemplative fade, mirrors an emotional arc of emergence, climax, and resignation.

Structurally the piece is through-composed: it opens with an intimate solo-piano introduction, adds synth pads, then layers strings and low brass as the dynamics rise toward a grand climax. After reaching the peak, the arrangement recedes, stripping back to piano and pads before fading out. This evolution is driven by dynamic contrast rather than repeated sections, reinforcing the piece's narrative flow.

Overall, the track embodies the modern neo-classical aesthetic that bridges concert-hall intimacy with film-score grandeur, situating itself within the contemporary "cinematic classical" movement that seeks to evoke visual-like storytelling through pure instrumental texture.

Figure 11: Example of a ground-truth caption on a classical instrumental piece in MF-Skills.

---

**Caption Evaluation Prompt: Cecilio G & Blackthoven - Life In Plastic (LqgegqS2QPE)**

Give me a full musical picture: what's the style, BPM, and harmonic palette, and how do the vocals, instrumentation, and production shape its mood?

**Output**

This track is an energetic Electro Latino / Latin Dance-Pop record that fuses bright, club-ready synth arpeggios with the rhythmic swagger of contemporary Latin urban music.

Tempo & Key – The song moves at ≈ 65.2 BPM and is anchored in C♯ minor , giving it a slightly darker harmonic foundation while the melodic content stays largely major-mode for a playful contrast.

Instrumentation & Production – The arrangement is built around crisp, punchy electronic drums (tight 808-style kicks, snappy snares, and rapid hi-hat rolls) that drive the groove. A synth bass delivers a thick, percussive low-end, while sparkling lead synths and arpeggiated pads fill the upper spectrum with a glossy, high-energy timbre. The mix is polished and modern: wide stereo imaging places the synth layers across the field, the drums sit forward with aggressive compression, and the overall mastering emphasizes loudness without sacrificing clarity. Reverb and delay are applied tastefully to create an expansive, club-ready soundstage.

Vocal Characteristics – A male vocalist (baritone/tenor range) alternates between rapid, rhythmic rap verses and melodic, autotuned singing in Spanish. His timbre is clear, confident, and slightly mischievous. Heavy pitch-correction, compression, reverb and delay give the voice a polished, almost synthetic sheen that matches the "plastic" aesthetic of the lyrics.

Lyrical Themes – The song satirically critiques hyper-commercialized beauty ideals, portraying a woman who is "artificial" or surgically enhanced. Repetitive chorus lines such as:

- " Siete mil en las tetas "
- " Barbie de silicona "
- " La chica de goma "

underscore the tongue-in-cheek commentary on material excess ("what's left in the Jordan"). The verses sprinkle pop-culture references (David Beckham, Black Eyed Peas, Katy Perry) to reinforce the theme of a fabricated, media-fed persona.

Structure & Dynamics – The track follows a conventional pop layout:

1. Intro – Atmospheric synth pads and a short spoken sample set the "Life in plastic" motif.
2. Verse 1 – Sparse beat with rap-style delivery, building tension.
3. Pre-chorus – Brief melodic hook that lifts the energy.
4. Chorus – Full-arrangement drop with layered synth leads, driving bass, and the catchy, autotuned vocal hook ("Siete mil en las tetas").
5. Verse 2 – Similar to the first verse, adding new lyrical details.
6. Bridge – Minimalist breakdown that isolates the vocal line before ramping back into the final chorus.
7. Final Chorus / Outro – Repetition of the hook with added vocal ad-libs and a fade-out of the "Life in plastic" chant, delivering a climactic, club-ready finish.

Dynamic variation is modest; the song relies on textural changes (adding/removing synth layers, shifting from rap to melodic sections) rather than large volume swings, maintaining a constant high-energy floor-filler vibe.

Overall Mood & Context – The piece radiates excitement, amusement, and a cheeky confidence, embodying the 2020s Latin-influenced EDM trend where producers blend reggaeton-flavored rhythms with European-style synth pop. Its satirical take on consumerist beauty culture aligns with a broader wave of Latin urban tracks that mix party-ready beats with socially aware—or at least self-aware—lyrics.

---

Figure 12: Example of a ground-truth caption on a Latin song in MF-Skills.

**Caption Evaluation Prompt: Aimless Crusade - 925 (PqYoMctVRco)**

Give me a full musical picture: what's the style, BPM, and harmonic palette, and how do the vocals, instrumentation, and production shape its mood?

**Output**

This track is an Indie Rock/Alternative Rock composition that blends earnest, guitar-driven songwriting with a slightly polished yet raw production aesthetic. It moves at a moderate 88 BPM in a steady 4/4 meter and is rooted in C♯ major, though the chordal landscape frequently dips into the relative minor (C♯ m) and the subdominant F♯ major family, giving the piece a bittersweet, introspective feel.

Instrumentation & production: The arrangement centers on clean and mildly overdriven electric guitars that are panned wide to create a spacious stereo field, a solid bass guitar anchoring the low end, and an acoustic drum kit providing a straightforward rock groove. Production is moderately polished: drums are tight with modest compression, guitars retain a touch of grit, and the mix places the male lead vocals front-and-center while preserving ambient room feel through light reverb and subtle delay. The overall mastering emphasizes clarity and dynamic range rather than a heavily compressed "wall of sound."

Vocal characteristics: Dylan Bond delivers baritone-to-tenor lead vocals with a clear, slightly strained timbre that conveys melancholy and anxiety. His delivery is earnest and melodic, staying in a clean register throughout; vocal processing is limited to light reverb, gentle compression, and a touch of delay, allowing the emotional nuance of the lyrics to remain prominent.

Lyrical themes: The lyrics explore feelings of overwhelm, regret, and emotional fatigue tied to a demanding work routine and a fading relationship. The recurring chorus line—"I'm working a 5 to 9, 9 to 5 / I think I'm working myself out of life… I still don't feel alright" —acts as a mantra of frustration, while verses hint at unrequited love ("I know that you want more from me… I'm sorry I could not be there for you"). The outro's instrumental fade underscores the lingering sense of unresolved tension.

Song structure & dynamics: The form follows a classic indie-rock template: Intro (guitar-based motif), Verse 1, Pre-chorus (implied by lyrical build), Chorus, Verse 2, Chorus (expanded with additional vocal lines), and an Instrumental Outro. Dynamics swell from the relatively restrained verses—driven by sparse guitar arpeggios and subdued drums—to the fuller, louder choruses where guitars open up, the bass thickens, and the vocal intensity peaks. The instrumental outro provides a gentle release, allowing the harmonic loop to resolve.

Theoretical grounding: The chord progression cycles mainly between the tonic C♯ major (I), the subdominant F♯ major (IV) and its first-inversion F♯/C♯, with occasional pivots to the relative minor C♯ m (vi) and a brief dominant-type Ab7 (♭VI7) that adds a blues-y tension before returning to the tonic. This alternation between major and minor tonalities reinforces the lyrical push-pull between hope ("maybe it'll be right again") and despair ("I don't know why I'm feeling this way").

Overall mood & aesthetic: The track exudes a melancholic, introspective mood characteristic of early-2020s indie-rock, channeling the anxiety of modern work life and relational strain through relatable lyricism and a melodic guitar-driven soundscape. Its blend of clean production with a raw edge situates it within the contemporary indie-rock tradition that values both emotional honesty and accessible songcraft.

Figure 13: Example of a ground-truth caption on an American song in MF-Skills.

**Caption Evaluation Prompt: Bené - Californie (2kudNV36plA)**

Give me a full musical picture: what's the style, BPM, and harmonic palette, and how do the vocals, instrumentation, and production shape its mood?

**Output**

This track is a polished French Trap / Melodic Rap piece that blends modern trap percussion with melodic, minor-key rap-songcraft, creating a confident yet slightly melancholy atmosphere.

Tempo & key – The song runs at ≈ 103 BPM in a 4/4 feel (time-signature not explicitly listed) and is rooted in A minor.

Instrumentation & production – The rhythm foundation is built from crisp electronic drums and booming 808 sub-bass, programmed with tight hi-hat rolls and snappy snares typical of contemporary trap. A deep synth bass underpins the groove, while a shimmering plucked synth carries the main melodic hook, complemented by lush, wide-stereo atmospheric pads that fill the background. The mix is high-fidelity: drums and bass sit centrally and punch through the mix, the melodic synths are spread wide, and the overall sound is polished with heavy use of auto-tune, reverb, and delay on the vocals, giving the track a spacious, glossy sheen.

Vocal characteristics – The lead is a male tenor whose timbre is smooth and heavily processed with auto-tune, delivering melodic rap that oscillates between sung phrasing and rhythmic rap flows. The French lyrics are delivered with a confident, boastful tone that can turn introspective, and the vocal chain adds prominent reverb and delay to enhance the hazy, modern trap aesthetic.

Lyrical themes – Although the full lyrics are unavailable, the described content centers on luxury and success (champagne, diamonds, helicopters), loyalty to friends, and overcoming challenges, with a romantic subplot that mixes affection and detachment. These themes are typical of French trap narratives that juxtapose material excess with personal struggle.

Song structure – The arrangement follows a classic verse-chorus format with a reflective bridge. An intro establishes the beat and plucked synth motif, leading into verses where the rap verses ride the trap groove. The chorus lifts the energy with a fuller harmonic texture and a more anthemic vocal hook, then returns to verses. A bridge provides a brief harmonic shift and lyrical pause before the final chorus and outro, which reinforce the track's main melodic motif.

Theoretical grounding – The chord progression cycles primarily through Am, Dm, and Em, with occasional Cmaj7, F6, and Bm7♭5/A extensions. This minor-centric movement (Am → Dm → Em) emphasizes a searching, slightly somber mood, while the inclusion of major-7 and sixth chords adds a subtle, lush coloration that softens the tension. The repeated return to the tonic Am in the choruses offers a sense of resolution and reinforces the track's emotive core.

Overall mood & aesthetic – The song projects a sleek, urban vibe that balances swagger with a hint of introspection, situating it firmly within the 2020s French trap scene where melodic rap and high-gloss production converge to portray both the allure and the inner conflicts of a lavish lifestyle.

Figure 14: Example of a ground-truth caption on a French song in MF-Skills.

**Caption Evaluation Prompt: Red Velvet - Eyes Locked, Hands Locked (Wn9MFo6tzJc)**

Write a rich caption that blends the technical details (genre, BPM, key, chords, mix) with how the song feels emotionally and dynamically as it unfolds.
Output the thinking process in <think> </think> and final answer in <answer> </answer>.

**Output**

<think>
Bright, midtempo synth-pop with a steady 4-beat pulse around the low-80s. The arrangement centers on shimmering pads and a bright arpeggiated lead spanning a wide stereo field, supported by a clean, lightly chorused electric guitar adding syncopated strums. Drums are electronic and tight: crisp snare, precise closed hats, and a deep, rounded synth bass anchoring the groove. The mix is glossy and spacious—vocals sit upfront and centered, synths are spread wide, and there's clear contrast between the intimate intro and the full, high-energy choruses.

Tonally it stays in a bright major center, cycling I–V–vi with ii and IV coloring; pre-choruses lean into vi to heighten tension before resolving back to I for the chorus lift. A brief minor iv inflection adds a bittersweet hue in the bridge before returning to the tonic.

Form is pop-standard: finger-snap intro → sparse verse with breathy delivery → layered pre-chorus build → big chorus drop with thick synth chords → verse 2 variation → repeat pre/chorus → stripped bridge with a more spoken cadence → short instrumental interlude reprising the arp → re-energized final chorus and chant-based outro fading on pads.

Female lead with a clear, youthful timbre: soft, slightly breathy verses contrasted by emotive belting in hooks; tight stacked harmonies, subtle melisma, light autotune polish, tasteful reverb, and firm compression. Lyrics are mainly Korean with English tags; the hook "눈 맞추고, 손 맞대고" recurs, framing themes of ecstatic first love, closeness, and cosmic blessing. Overall mood: joyful, romantic, and radiantly polished K-Pop.
</think>
<answer>
This track is a bright K-Pop song that leans heavily into synth-pop, blending polished electronic textures with the melodic sensibility of contemporary Korean pop. It moves at ≈ 81 BPM, is rooted in B♭ major, and (though the time signature isn't listed) follows a standard 4-beat pulse typical of mainstream K-Pop.

Instrumentation & production – The arrangement is built around shimmering synth pads and arpeggiated synth leads that sweep across a wide stereo field, a clean electric guitar that adds occasional rhythmic strums, and a punchy electronic drum kit with crisp snares, tight hi-hats, and a deep, rounded synth bass that drives the low end. The mix is high-fidelity and glossy: vocals sit front-center, synth layers are spread wide, and the mastering emphasizes clarity and loudness while retaining dynamic contrast between the gentle intro and the high-energy choruses.

Vocal characteristics – A female lead delivers the melody with a clear, bright, youthful timbre. In the verses her voice is soft and slightly breathy, creating intimacy; during pre-choruses and choruses she shifts to powerful, emotive belting, often layered with tight harmonies. The vocal style is melodic and melismatic, typical of K-Pop, with light autotune for pitch polish, subtle reverb for space, and compression for consistent presence. Lyrics are primarily Korean with English tags such as "Oh yeah" and "Could you begin our strange love?".

Lyrical themes – The song celebrates an ecstatic, first-love connection, repeatedly urging the listener to "눈 맞추고, 손 맞대고" ("look into each other's eyes, hold hands"). The chorus repeats the hook "우리의 마음 마음 닿을 때 … 빠져버렸어 yeah" ("When our hearts touch, I'm lost in your thoughts, yeah"), underscoring a theme of being completely captivated by the beloved. Verses paint a shy morning meeting, while the bridge expands the romance to cosmic imagery ("내가 아는 우주도 우리를 축복해").

Structure & dynamics –
- Intro – Finger-snapped synth motif establishes a gentle ambience.
- Verse 1 – Sparse accompaniment, breathy vocal delivery.
- Pre-Chorus – Builds with added percussion and layered synths; vocal intensity rises.
- Chorus – Full-arrangement drop with driving drums, thick synth chords, and belted vocal hooks; dynamic peak.
- Verse 2 – Mirrors the first verse with slight melodic variation.
- Pre-Chorus / Chorus – Reiteration of the earlier build, reinforcing the hook.
- Bridge – Stripped-down synth pads and a more spoken-like vocal line, creating a brief lyrical pause before the final surge.
- Interlude – Short instrumental break that re-introduces the arpeggiated motif.
- Final Chorus & Outro – Re-energized chorus followed by a repetitive outro of the "눈 맞추고, 손 맞대고" chant, fading out with the synth pad texture.

Theoretical grounding – The chord progression cycles mainly through the tonic B♭ major (I), dominant F (V), and sub-mediant Gm (vi), with occasional ii (Cm7) and iv (Eb) chords that keep the harmony firmly in a major, uplifting mode. This I–V–vi movement creates a sense of forward motion and emotional release, especially when the pre-chorus shifts toward the vi (Gm) before resolving back to the bright I (B♭) in the chorus.

Overall mood & context – Radiating joy, romance, and youthful optimism, the track exemplifies the modern K-Pop aesthetic of 2020s global pop: high-gloss production, hook-laden choruses, and a seamless blend of electronic synth-pop with Korean melodic phrasing. It situates itself within the current wave of K-Pop songs that pair infectious dance rhythms with heartfelt love lyrics, aiming for both domestic chart success and international streaming appeal.
</answer>

Figure 15: Example of a ground-truth caption with thinking traces on a Korean song in MF-Think.

**Caption Evaluation Prompt: Ricky Bishop - Anlè Mwen (9_Ktzlc0OvU)**

Write a rich caption that blends the technical details (genre, BPM, key, chords, mix) with how the song feels emotionally and dynamically as it unfolds.
Output the thinking process in <think> </think> and final answer in <answer> </answer>.

**Output**

<think>
A shouted chant kicks things off, instantly setting a party-call vibe before the groove locks in. The beat feels fast and danceable, like double-time trap, but with a Caribbean sway—congas and shakers add a lilt that pushes the offbeats and gives it a Kompa-style swing. The kick is tight and punchy, glued to a deep sub/808 that drives the low end with clean, sustained notes; snares/claps snap on the backbeat while crisp hats sprinkle in light rolls and occasional triplet stutters.

Harmonically it sits in a dark minor center; the bass keeps returning to a low tonic while the keys/patches color the harmony with tension—there are moments that sound like half-diminished or altered leading-tone chords before resolving. Every so often a brighter major color lifts the mood briefly (A/E-flavored sheen) before dropping back into the moody minor.

The vocal is a confident male baritone rapping in Creole with French slips, upfront and tightly compressed. Heavy Auto-Tune gives it a melodic edge. Throughout, chopped and pitch-shifted vocal bits answer the lead, drenched in delays and big reverbs that bloom in the sides—classic call-and-response hooks.

Structure is hook-driven: chanty intro → hard-hitting verse over the syncopated percussion → a chorus that layers extra ad-libs and crowd-style shouts → a more melodic bridge where the delivery softens into a sing-rap feel → final chorus and a cyclical return to the opening chant. The overall mood is bold, celebratory, and club-ready.
</think>
<answer>
This track is an energetic Haitian Rap piece that fuses the dance-floor swing of Kompa with contemporary trap-style hip-hop, creating a high-energy "Kompa-Rap" blend. It moves at 150 BPM and is rooted in C♯ minor , giving it a dark tonal center while the rhythmic drive feels uplifting.

Instrumentation & production: The mix is polished and punchy, built around a deep, rhythmic 808-type synth bass that locks with a layered electronic drum kit and live-sounding percussive elements such as congas and shakers. Sparse synth accents punctuate the groove, and the percussive tracks are spread wide in the stereo field for a dynamic, club-ready feel. The overall production is clean, with tight compression on the drums, crisp high-frequency detailing on the synths, and a balanced low-end that lets the bass pulse through without muddying the vocals.

Vocal characteristics: The lead is a male baritone delivering confident, assertive rap verses in Haitian Creole. His timbre is clear and slightly aggressive, heavily processed with Auto-Tune for a melodic-rap texture. Additional vocal samples are pitch-shifted, chopped, and drenched in reverb and delay, adding rhythmic hooks and call-and-response elements throughout the track.

Lyrical themes: The lyrics celebrate street credibility, personal success, and defiant pride. Refrains such as "Kòman yo vle mwen, kòman yo vle mwen" and "se de l'histoire ancienne" underscore a boastful narrative about overcoming opposition and owning one's reputation. The chorus repeatedly chants "Konbyen fanm ka … an scred" , reinforcing the track's hook-driven, chant-like quality.

Song structure: The arrangement follows a classic intro-verse-chorus-bridge-outro format. An opening chant of "Anlè Mwen" establishes the hook, leading into verses that stack rapid rap flows over the syncopated Kompa-inspired percussion. The chorus expands the hook with layered vocal samples, then a bridge (or "Pont") introduces a more melodic, chant-like passage before the final chorus and an outro that repeats the introductory chant, bringing the piece full circle.

Harmonic analysis: Although the track sits in C♯ minor, the chord progression frequently moves through related minor-seventh and half-diminished chords (e.g., F♯m7♭5, Abm7♭5, B7/D♯) before resolving to the tonic C♯m or the relative major Abm7. This shifting palette creates tension—especially the use of diminished-type chords—while the recurring return to C♯m and the bright Amaj7 moments provide release, mirroring the lyrical confidence.

Overall mood & cultural context: The song radiates a bold, celebratory mood, marrying the rhythmic vitality of Haitian Kompa with the modern swagger of trap-influenced rap. Its use of Creole lyrics, Caribbean percussion, and contemporary Auto-Tune aesthetics situates it firmly within the current wave of Haitian urban music that honors traditional dance rhythms while pushing into global hip-hop territory.
</answer>

Figure 16: Example of a ground-truth caption with thinking traces on a Haitian song in MF-Think.

---

**Caption Evaluation Prompt: Nikn - Abgrund (KDBvKOQLRrY)**

Write a rich caption that blends the technical details (genre, BPM, key, chords, mix) with how the song feels emotionally and dynamically as it unfolds.
Output the thinking process in <think> </think> and final answer in <answer> </answer>.

**Output**

<think>
A low, brooding pad and filtered sub set a tense scene before a sudden drop into a tight 4/4 trap bounce, roughly 120 BPM. The drum programming is classic and punchy: razor-edged closed hats with quick 1/16 rolls, crisp snare cracks on 2 and 4, and an 808 that slides and occasionally saturates into gritty distortion. The bass and kick feel glued, with sidechain breathing the pads subtly.

Tonally it sits in a dark minor center—Fm is the ear's anchor. I keep hearing a loop that lifts out of the gloom with glossy major-seventh color (Abmaj7, then Dbmaj7), a bright sheen against the otherwise shadowy palette. In the bridge there's a tense half-diminished color over F and a slash-chord lift that suspends gravity before dropping back to the tonic or Eb—nice push-pull. Near the end, the harmony thins and hints at a modal sidestep as the texture strips down.

The vocal is a male rapper in Swiss dialect, forward and aggressive, with light auto-tune and short, controlled reverb—modern but not overprocessed. The hook works like a mantra, built on a defiant chant that sits squarely in the pocket. A brief bridge brings a semi-sung, more melodic tone, then the verses snap back with percussive attack.

In the outro the beat recedes to pads and a restrained sub while the voice switches to dry, intimate spoken word in standard German—philosophical, minimal, almost abstract-hip-hop in its austerity. The overall arc: high-intensity, bass-driven swagger that opens space at the end for reflection.
</think>
<answer>
This track is an aggressive Swiss-German trap piece that leans into abstract-hip-hop on its closing spoken-word section, creating a stylistic blend of hard-hitting trap beats with a minimalist, experimental hip-hop outro.

Tempo & Meter – The song moves at 120 BPM in a steady 4/4 time-signature, anchored by a booming, often distorted 808 bass that locks in with crisp, punchy trap-drum patterns (tight hi-hats, snappy snares, and deep sub-kicks).

Key & Harmony – The tonal centre is F minor . Although the harmonic content is sparse, the chord list reveals a recurring progression that emphasizes the minor mode while occasionally brightening the texture with its relative major and borrowed chords:

- Verses and pre-choruses hover around Fm7 → Abmaj7 → C♯maj7 , a loop that keeps the mood dark yet slightly lifted by the major-seventh colour of Ab.
- The bridge introduces Gm7♭5/F and C♯/G♯ , adding tension before resolving back to Fm7 or Eb , reinforcing the track's searching, defiant character.
- The outro leans on Abm6 and Dm , hinting at a modal shift that underscores the philosophical spoken word.

These progressions create a sense of cyclical tension (minor-side emphasis) that resolves in the choruses, where the harmonic movement is minimal and the focus shifts to rhythm and vocal delivery.

Instrumentation & Production –
- Low end: An extremely prominent, heavily saturated 808 that often bends into distortion, providing the track's aggressive backbone.
- Drums: A classic trap kit—tight, high-frequency hi-hats, crisp snares, and deep sub-kicks—processed with aggressive compression and transient shaping to cut through the mix.
- Synths: Dark, atmospheric pads fill the stereo field, while sparse, repetitive synth melodies (often single-note motifs) punctuate the verses. Occasional sound-effects (whooshes, risers) accentuate drops and transitions.
- Mixing: High-fidelity yet raw; the mix centers the vocals, spreads drums and bass wide, and uses subtle side-chain pumping on the pads to keep the low end tight. Reverb and a modest amount of auto-tune on the rap vocals add a modern trap sheen without sacrificing clarity.

Vocal Characteristics –
- Male rap in Swiss German dominates the verses: a fast, confident, aggressive flow with a clear, direct timbre. Light auto-tune and short-reverb tails give a contemporary trap polish.
- Melodic bridges introduce a slightly sung, more melodic tone, providing a brief tonal contrast before returning to the rap intensity.
- Outro spoken word (4:04-4:21) shifts to standard German, delivered in a somber, reflective style, stripped of effects to highlight the philosophical lyrics.

Lyrical Themes – The lyrics revolve around defiance, self-assertion, and a critical view of society, punctuated by a rallying chorus that repeats as a mantra of confidence:

- *"Mir si hie am Tanze, ei fuess immer über der kante"* (We're dancing here, our feet always over the edge)
- *"Chum, chum mir springe, ey, mit Flueg gwünne King, ey, dreck."* (Come, come let's jump, with wings we'll win, king, damn)

These repetitive lines act as a call to action, reinforcing the track's aggressive, unapologetic stance.

Structure & Dynamics –
1. Atmospheric intro – Dark synth pads and a filtered 808 set a tense mood.
2. Verse 1 – Full-force trap beat with rapid Swiss-German rap.
3. Chorus – Repetitive hook with the defiant chant, drums lock into a tighter groove, bass stays dominant.
4. Bridge – Slightly melodic, softer synth texture, vocal tone becomes more sung, providing a brief dynamic dip.
5. Verse 2 / Verse 3 – Return to high-energy rap, maintaining intensity.
6. Final Chorus – Reinforces the mantra, with subtle layering of vocal harmonies for added impact.
7. Outro (spoken word) – Beat strips back to minimal pads and a subdued 808, allowing the philosophical German monologue to close the piece on a reflective note.

Overall Mood & Context – The track exudes a fierce, confident, and defiant aura, embodying the contemporary Swiss-German trap scene where regional dialects meet global trap production aesthetics. Its raw, bass-heavy soundscape, combined with the occasional abstract-hip-hop outro, positions it within the 2020s wave of European trap that values both street-level swagger and introspective lyrical depth.
</answer>

Figure 17: Example of a ground-truth caption with thinking traces on a European song in MF-Think.

**Caption Evaluation Prompt: Tears For Fears - Head Over Heels (CsHiG-43Fzg)**

```
Which label best captures the track's core style and sound palette?
Choose from the following options
A. Alternative/Indie Rock with atmospheric synth undercurrents
B. Synthwave/Retrowave instrumental
C. Acoustic folk ballad with minimal backing
D. Heavy metal with double-kick and chugging riffs
E. Dream-pop/shoegaze with washed-out vocals
F. Pop-punk with brisk, palm-muted power chords
G. New Orleans traditional jazz ensemble
Output the thinking process in <think> </think> and final answer in <answer> </answer>.
```

**Output**

```
<think>The backbone is live rock: steady kit, melodic electric bass, and guitar-led writing.
Verses feature clean, arpeggiated electric guitar; choruses thicken with layered, mildly
distorted rhythm guitars. Spacious pads swell behind the band, but the song's feel, form, and
drum/bass groove are rock rather than electronic. The synths color the atmosphere rather than
drive the beat, placing it in alternative/indie rock with synth accents.</think>
Hence, the final answer is: <answer>Alternative/Indie Rock with atmospheric synth
undercurrents</answer>
```

Figure 18: Example of QA pairs with thinking traces in MF-Think.

**Caption Evaluation Prompt: Emel Sayın - Gidenler Dönmez (oTd-YQ51f9w)**

```
Right before the first full chorus with the line "Gidenler dönmez" (around 1:30), what chord
does the band land on to set up the hook?
Choose from the following options
A. Gm
B. D
C. Cm
D. F
E. Bb
F. Eb
G. A
H. E
Output the thinking process in <think> </think> and final answer in <answer> </answer>.
```

**Output**

```
<think>Just before the hook, the harmony tightens on the dominant: the bass centers on D, the
guitars outline a major-quality V, and there's a classic build that resolves into the minor
tonic under the chorus. That dominant prep is D.</think>
Hence, the final answer is: <answer>D</answer>
```

Figure 19: Example of QA pairs with thinking traces in MF-Think.

**Caption Evaluation Prompt: ionnalee - play in KONSERT (vaSL5P9DCgw)**

```
How would you characterize the drum and bass production?
Choose from the following options
A. Metallic, rigid, and heavily compressed; distorted bass central and grinding
B. Loose live kit with roomy ambience; warm upright bass
C. 808 trap with rolling hi-hat triplets and sub-only bass
D. Breakbeat/jungle with chopped amen breaks
E. Acoustic percussion with hand drums and shaker
F. Glitch-hop swing with lopsided grooves
Output the thinking process in <think> </think> and final answer in <answer> </answer>.
```

**Output**

```
<think>The snare is crisp and industrial, the kick punchy and tight; bass surges with
saturation—all locked to a machine-like grid.</think>
Hence, the final answer is: <answer>Metallic, rigid, and heavily compressed; distorted bass
central and grinding</answer>
```

Figure 20: Example of QA pairs with thinking traces in MF-Think.

**MF-Skills Detailed Caption Generation Prompt**

You are a music expert and are given the task to generate accurate music captions from the metadata provided to you. I will provide you with several types of metadata about music. These metadata include:

1) Synopsis: a richly detailed global description covering various details about the input music, including genre, sub-genre, etc
2) Lyrics: Lyrics for the song (possibly timestamped -- if timestamped, will be [start, end, lyrics]). Sometimes lyrics will be in parts, which would mean the music is sourced from a file that has multiple songs.
3) Ground Truth Tempo: The average tempo of the song
4) Ground Truth Key: The ground truth key for the song.
5) Duration of the song
6) Chord Poregression

Generate a caption with the following constraints:
1) The caption should have the following details:
a) Clearly state the genre, sub-genre, and stylistic blend of the track.
b) Include the tempo (BPM), time signature, and key as provided in the metadata (**please do not use details in the synopsis or other places for this info, only use the info separately provided to you**). Describe the instrumentation and production choices in detail (e.g., guitars, synths, drums, mixing/mastering style).
c) Summarize the vocal characteristics, covering timbre, gender, vocal style (clean, screamed, etc.), and effects.
d) Capture the lyrical themes with illustrative examples if lyrics are available. Include repetitive lyrics, like the chorus, etc.
e) Highlight the song structure (intro, chorus, breakdown, bridge, finale) and how it evolves dynamically.
f) Optionally, if you have enough knowledge of music theory, enrich the explanation of perceived structures with theoretical grounding (e.g., how verse progressions emphasize the minor side, such as Em to Bm to create a searching mood, resolving into choruses centered around G, C, and D for emotional release). If not possible, ignore this point.
g) Convey the overall mood and aesthetic, situating the track within its cultural or historical context if relevant.
2) If some metadata is not available, you will be provided with \"Not Available\". In such a case, just ignore having that information in the caption, but do not put wrong information.

An example caption is: \"This track is a high-energy Electronicore piece blending trance synths with metalcore aggression. At ~178 BPM in 4/4, primarily in E minor, it drives forward with relentless double-kick drums, distorted power-chord guitars, and pulsating arpeggiated synthesizers. The harmonic foundation is built around E minor-centered progressions, often revolving between E minor, C major, and D major chords, with occasional shifts toward G major to brighten the mood. These cycles create a sense of tension and release, reinforcing the track's melancholic yet anthemic character. Instrumentation includes deep synthesized bass, bright electronic pads, soaring lead synths, and rhythm guitars layered into a compressed 'wall of sound' production. Vocals are diverse: a melodic female soprano delivers passionate French lyrics with reverb and delay, a clean male tenor provides anthemic harmonies, and harsh male screams add raw intensity in breakdowns. Lyrics explore themes of pain, resilience, and escape, with lines like 'Demain je pars sur Mars' ('Tomorrow I leave for Mars'). Structurally, the song evolves from a contemplative synth intro over a descending minor progression (Em-C-D) into explosive choruses featuring rising cycles (Em-G-D-C), breakdowns anchored on sustained pedal E, a spoken-word bridge, and a climactic finale. Production choices emphasize wide stereo imaging, bright EQ, and aggressive compression, ensuring drums and leads cut sharply through. Overall, the track combines electronic euphoria with metal ferocity, embodying the 2010s French Trancecore movement.\"

**Output only the generated caption and nothing else**. Here is the input information:

Figure 21: Prompt for generating detailed captions for the MF-Skills dataset.

---

**MF-Skills QA Generation Prompt**

"You will receive N audio files with per-file METADATA blocks. For EACH file i (i = 1..N), produce a diverse set of expert-level Q&A pairs grounded ONLY in the audio and the metadata.

Return ONE valid JSON object. Its top-level keys must be "1","2",… in the same order as inputs. Each value is an object with:
"path": string (unchanged, the file's unique path/id),
"qna": [ {{ "question": string, "type": "open" | "multi-choice" | "count" | "time-based" | "chord" | "technique" | "comparison", "options": [string]?   # include only for multi-choice; 3-5 plausible options
"answer": string,      # exact/concise answer
"rationale": string,   # brief justification grounded in audio/metadata
"references": {{ "time_seconds": [float]?
# timestamps referenced (e.g., where the event occurs) "bars": [int]? # if bar numbers are referenced
"chords": [string]?# if chords are referenced }} }}, ... ] ]

Constraints & guidance:
• Base every fact on the provided audio and metadata; avoid speculation.
• Prefer concrete, checkable questions (timings, instrument onsets, entry of vocals, chord names, count of voices/instruments).
• When metadata includes BPM, key, or chord labels, you may use them but verify against what is audible; if conflict, say "according to metadata".
• Use exact timestamps when the question depends on a moment or interval.
• Include several question types and difficulty levels. Aim for 12-20 Q&As per file.

• Examples of acceptable question styles (adapt/adopt as appropriate; do not copy verbatim):

- Time-based: "When does the singer begin?" - Mid-section event: "Midway through, which instrument solos?"
- Counting: "How many total voices are singing?" - Segmentation: "Count the number of songs in the recording."
- Contrast/MCQ: "What changed midway? A: key B: singer C: tempo D: background instrument"
- Lyric-localization MCQ using supplied lyric text. - Chord-localization: "What is the root chord from 0.00 to 1.60?"
- Chord-identity MCQ for a given interval. - Last chord + duration. - Technique & orchestration (e.g., spiccato, rubato) with bar/time references.
- Interval range (lowest↔highest pitch).
- II-V-I identification MCQ.
- Comparison between two audio segments within the same file (if clearly distinct).
- Chorus start time: "When does the first chorus begin?" - Drop timing: "At what timestamp does the drop land?"
- Tempo & meter ID (MCQ): "What is the meter? 4/4, 3/4, 6/8, mixed"; "Does the tempo change after 1:00? If yes, when?"
- Key modulation: "When is the first key change, and from which key to which?"
- Groove shift: "When does the groove switch to half-time/double-time?"
- Lyric trigger: "At what time does the word '___' first appear?" / "Which line directly precedes the chorus?"
- Lead role (late section): "What is the lead instrument in the outro/latter part?" - Instrument entries/exits: "Which instrument enters first after the drop (timestamp)?"
- Drum/motif counts: "How many drum fills occur after 1:00?" / "How many repeats of the main motif?" - Section durations: "How long is the bridge (in seconds/bars)?"
- Preceding chord: "Which chord is immediately before E:min9(*5)/1?"
- Progression detail: "What is the fifth chord in the progression?"
- Cadence type: "HC / DC / PAC / IAC"
- Harmony under expression: "What is the harmony when rubato occurs? (FM13#11 / CM7#11 / Bm7b5 / EbM9)"
- II-V-I mapping (names MCQ): "Am11, D13b9, Cmaj7add13" … (four options)
- Melody specifics: "Where does the main motif repeat verbatim? (timestamp)" / "Identify one blue-note occurrence."
- Rhythm features: "Give one timestamp with clear syncopation." / "Is there a polyrhythm or hemiola? Where?"
- Bassline function: "Is the bass mostly root-motion, walking, or pedal? Justify with times." - Inversion spotting: "Identify an inversion (e.g., I⁶) and its time."
- Production/mixing: "Which element is hard-panned left/right?" / "Is sidechain pumping audible? Provide a timestamp."
- Space & FX: "What reverb character is on the vocal (plate/room/hall)? First audible tail time?" - Dynamics: "Where is peak loudness?" / "Mark a clear crescendo span (start-end)."
- Notation details: "In which bar does the guitar first use palm-mute?" / "What are the left-hand piano notes in bar ___?"
- Similarity/difference MCQ: "What is NOT a reason these two versions sound familiar? (singer/rhythm/key/tempo)"
- Orchestration comparison: "What improvements were made in orchestration between segments?"
• **There is no need to generate all types of questions for every audio. Rather, generate only the questions that are most suitable.**
• **Use the metadata provided as a reference, but also understand the audio on your own.**
• Do not include things like "according to the metadata", or anything similar in your question. Assume the model I will train with this data will only get the song as input.
• Output ONLY the JSON object. No extra commentary. Now I will provide the files and their METADATA blocks in order:

Figure 22: Prompt for generating QA pairs for the MF-Skills dataset.

---

**Caption Re-writing Prompt (with lyrics)**

I will provide you with a music caption. Together with the caption, I will provide you with lyrics (possibly time-stamped and segmented). You need to rewrite the caption with lyrics integration. An example is:

Original Caption: This upbeat blues-rock track features lively guitar riffs and rhythmic piano, supported by driving drums at a brisk tempo of around 93.75 BPM. The music blends rockabilly and country influences with expressive, sparse vocals conveying a nostalgic, longing mood.

Rewritten Caption: This upbeat blues-rock track features lively guitar riffs and rhythmic piano, supported by driving drums at a brisk tempo around 93.75 BPM. The music blends rockabilly and country influences with expressive, sparse vocals that carry themes of loss and departure — with lines recalling mama's warnings, farewells to a lover, and the pain of moving on ("I'll be gone, must be gone"). The performance conveys a nostalgic, longing mood while maintaining spirited energy.

In multiple cases, the transcript will be short or not make sense, or integrating with the original caption will not lead to a good final output caption. In such a case, do nothing and just return the original caption. Only return the final caption and nothing else.

Here are the input details:

---

Figure 23: Prompt for correcting existing captions with lyrics and metadata on Music4ALL and MSD datasets.

---

**QA Correction Prompt**

You are an audio expert. I will provide you with a question about an audio sample (maybe music, sounds, or speech), a list of options, and the correct option. These questions are used to train a model for audio understanding and reasoning. Your main task is to make the data (question + option combination) more difficult, so that training on the data can provide better learning signals. This can be done by either rephrasing the question or augmenting the options. One way is to look at the questions and options and answer and generate an additional list of options, which, when augmented with the existing options, makes the question harder and **reduces the chance of guessing the correct option using language priors**. This can be done by generating confounding options similar to the correct option and different from existing.

Here is one example:
Input Question: Which scenario would this piece be most suitable for?
Input Options:
(A): Horror film suspense scene
(B): Science fiction video game background
(C): Romantic comedy montage
(D): Live jazz club performance
Correct Option: Science fiction video game background

Additional generated options to make it harder: ["Futuristic film trailer cold open", "Esports arena hype stinger", "Tech product launch highlight reel", "Noir detective game menu screen", "Cyber-mystery series interstitial", "Post-apocalyptic series teaser"]

Reasoning: Only one option in the question maps cleanly to electronic/futuristic tropes; distractors are caricatures. So additional options make it harder.

Finally, if you feel the question itself has a language prior, rephrase the question itself.\n\nIf you are generating new options, generate anywhere between 3 and 8 additional options, as you deem fit, to the point where you feel the question is already difficult. Not all questions, options, and answers may sound like they have priors. So in this case, feel free to augment any additional options you deem fit. **Do not just use the explicit information provided, do reasoning and think hard about implicit factors.**

Generate a new question, new options, or both. Only output a JSON with 3 keys: {"Question": The original question or corrected one., "Options": [list of strings with Options including the original options], "Answer": Just the original answer.}

Here is the input information:

---

Figure 24: Prompt for option augmentation of existing question-answer pairs on the Music4ALL and MSD datasets.

**MF-Think Music CoT Generation Prompt**

You are a music expert. I will provide you with a music caption that provides a detailed description of a song/music sample in an abstract and concise fashion. I will use this caption for training a model for music understanding and captioning, where the model gets just the input song and is expected to generate the caption. Additionally, I will provide you with extra ground-truth metadata about the music, like keys, chords, bpm, lyrics, etc. You need to generate a thinking trace that a human music expert would go through for generating such a caption.

Objective: The final goal is to train a model to think before generating the provided caption, given only the input song. The thinking trace should therefore be written as if listening to the track directly, not using metadata. Metadata is provided only to help you avoid mistakes and understand the song to better reflect how a musician would understand and think about it, but should not be mentioned in the reasoning.

Instructions:\n- Generate a natural, coherent thinking trace that reflects how a human expert would analyze the track to arrive at the caption.
- Focus on key aspects: tempo/feel, rhythm and groove, instrumentation and production, tonality (key, chords, harmonic color), vocals (timbre, style, processing), lyrics and themes, structure, mood, and genre context, using both the caption and the metadata.
- Style should follow the sample provided (natural, descriptive, step-by-step reasoning).\n- Do not restate the caption; only provide the reasoning that could lead to it.
- Length: less than 250 words.
- Output only the thinking trace, nothing else.

Example (for style):\n"The track opens with a phone-message sample, setting a gritty, streetwise mood before the beat drops. The tempo feels fast for trap, around 130 BPM in 4/4, giving it a driving, club-ready pulse. The production is anchored by booming 808s, razor snare cracks, and stuttering hi-hats, while polished mixing adds wide stereo pads and risers behind the drums.\nTonally, the piece centers on E minor. I hear Em7 as the tonic, but also bright chords like Amaj7, Bbmaj7, and Fmaj7 that briefly shift the mood upward before returning to minor, creating a push-pull tension that gives the track a glossy sheen beyond typical trap minimalism.\nThe vocals are male baritone, rapped aggressively in German. Processing is heavy — autotune, compression, and subtle delay/reverb keep the delivery confident and upfront. The hook relies on a chant ("Waynak, waynak? Alo, alo"), repetitive and catchy, clearly designed as an earworm for club settings.\nLyrical content centers on braggadocio, luxury, speed, and dominance, rooted in German street swagger. The structure cycles through intro → hook → verses → pre-hooks → repeated hook → outro, reinforcing the chant as the main motif.\nOverall: a boastful, polished German trap anthem with heavy low-end drive, minor tonality colored by glossy chords, chant-based hooks, and high-energy club production. Taken together, these elements define the track's overall character and set the stage for a concise description; with this analysis in mind, I can now capture the essence of the track in a single caption."

Here is the input information:\n\n

Figure 25: Prompt for generating step-by-step reasoning for the detailed captions in the MF-Think dataset.

QA CoT Generation Prompt

You are a music expert. I will provide you with a set of Q&As, which consists of questions, answers, and optionally options about a song/music sample. I will use this QA for training a model for music understanding and reasoning, where the model gets just the input song and is expected to generate the correct answer. Additionally, I will provide you with a caption and extra ground-truth metadata about the music, like keys, chords, bpm, lyrics, etc. You need to generate a thinking trace that a human music expert would go through for generating such a caption.

Objective: The final goal is to train a model to think before generating the provided answer, given only the input song. The thinking trace should therefore be written as if listening to the track directly, not using metadata. Metadata is provided only to help you avoid mistakes and understand the song to better reflect how a musician would understand and think about it, but should not be mentioned in the reasoning.

Instructions:
- Generate a natural, coherent thinking trace that reflects how a human expert would analyze the track to arrive at the answer.
- Focus on key aspects: tempo/feel, rhythm and groove, instrumentation and production, tonality (key, chords, harmonic color), vocals (timbre, style, processing), lyrics and themes, structure, mood, and genre context, using both the caption and the metadata.
- Style should follow the examples provided (natural, descriptive, step-by-step reasoning).

Additionally, sometimes the question and the answer I would provide refer to the metadata (like "The track's tempo is described differently in two places. What tempo is stated in the descriptive text, and what BPM is listed elsewhere?"). In the case, please rephrase the question, answer, and optionally options ("What is the BPM of the song?" —> answer from the metadata —> you need to look at the answer to rephrase the Q&As for these). If the question is confusing and does not make sense for a model that only receives the song as input, don't return anything for the QA. Also, some questions have language priors in the question. In that case, your main task is to make the data (question + option combination) more difficult, so that training on the data can provide better learning signals. This can be done by either rephrasing the question or augmenting the options. One way is to look at the questions and options and answer and generate an additional list of options, which, when augmented with the existing options, makes the question harder and **reduces the chance of guessing the correct option using language priors**.
**Finally, some questions that ask for things like key and BPM may be too simple and not require thinking. In that case, don't return anything for the question or return a short thinking trace, as you deem fit.\n- Length: less than 250 words.

Example Thinking:
"question": "How would you characterize the track's production style and overall mix quality?", "answer": "Lo-fi, sample-based production that is nonetheless well-mixed.", "thinking_trace": "The drums sound like chopped breaks—tight kick, snappy snare—treated with light saturation. The melodic bed is a soulful loop with slight vinyl/tape texture. Most elements feel mono-centric with modest stereo on the sample tail, yielding an underground, gritty character. Despite the lo-fi aesthetic, balances are tidy: vocal sits neatly above the loop, lows are controlled, and transient definition remains clear—so it's lo-fi by design yet well-mixed." --> this is a small thinking chain, feel free to go up to 250 words, but remember to not put irrelevant thinking points in the traces.\n\nOutput a JSON with the subjoins for each QA, where each sub-json has question (possibly rephrased), answer, and options (possibly augmented), and the **reasoning trace**. like {"qa1": {"question": …., "answer": … "options": .. "thinking trace": ...}, "qa2"}

Here is the input information:

Figure 26: Prompt for generating step-by-step reasoning for the question-answer pairs in the MF-Think dataset.

