# OpenReview forum: "Music Flamingo: Scaling Music Understanding in Audio Language Models"
_ICLR.cc/2026/Conference — ICLR 2026 Poster_

### Official Review · Reviewer_qEKD · 2025-10-27

**Soundness:** 3
**Presentation:** 3
**Contribution:** 3
**Rating:** 6
**Confidence:** 3

**Summary:**

This paper proposes Music Flamingo, a large audio–language model, designed to advance music understanding in foundational audio models. Music Flamingo is a fine-tuned model based on Audio Flamingo 3 backbone. This paper also introduces MF-Skills and MF-Think, two large-scale datasets containing music–caption and music–QA pairs designed to promote deliberate reasoning.

**Strengths:**

-	Music Flamingo achieves state-of-the-art results on 12 music understanding and reasoning benchmarks.
-	Low-level attributes are extracted using the MIR tool, and initial captions combined with these attributes are used to reduce hallucination, creating captions and QA data that include multifaceted information to construct MF-Skills data. Synthetic data is built with attention to hallucination by LLM, considering important characteristics in the music domain, and an effective approach has been proposed.

**Weaknesses:**

-	The ablation study is missing. Although improvements are made based on AudioFramingo 3, the model is trained in three stages: w/MF-skill, w/MF-think, and GRPO with custom rewards. Having evaluation results for each stage would help confirm each contribution.
-	Some details about the dataset creation are not described.

**Questions:**

- There seems no explanation of how and from where the 2M songs for MT-skills were collected. Additionally, is there a possibility that the data used for evaluation is included in the training data?
- I could not find an explanation for why the number of examples increases from 2M songs to 3M examples in MT-Skills. Does this mean that captions are added to all 2M songs, and some songs also have QA, thus resulting in 3M examples including pairs of music-caption and/or music-QA?
- Does AF3-St. 3 in Table 2 refer to the training data of the base model AudioFramingo3? Additionally, does 2.0 of MF-SFT mean adding MF-Skills twice? Are the other data also used in the training of AudioFramingo3 and then reused in fine-tuning for Music Framingo?
- It would be beneficial to elaborate on how quality filtering is performed.
- How were the categories in Figure 27 analyzed?
- Will the created dataset and source code/checkpoints be made publicly available?
- How were the inputs and outputs for ACC evaluation handled?
- In Figure 3, the term "MF-skills CoT" is used, but the caption seems to explain it as "MF-Think." It would be better to standardize the terminology.
- It seems Figures 4 and 27 are identical, but only Figure 27 is mentioned in the text. It would be better to refer to Figure 4 and note in its caption that an enlarged version is included as Figure 27 in the Appendix.

---

> ### Author Response · Authors · 2025-11-19
> **Response to Official Review of Submission7878 by Reviewer qEKD (1/2)**
>
> We thank you for your thorough review and constructive feedback. We're grateful for your recommendation to accept our paper. We have tried to address each of your concerns point by point.
>
> ------
> ### Questions:
>
> > There seems no explanation of how and from where the 2M songs for MF-Skills were collected.
>
> **Ans.** All songs in MF-Skills were downloaded from publicly available internet sources and ethically sourced. As part of our open-source release, we will provide links to the audio along with the corresponding labels (captions, QA pairs, metadata, etc.).
>
> > Additionally, is there a possibility that the data used for evaluation is included in the training data?
>
> **Ans.** No. We ensured that all evaluation sets are fully disjoint from the training data. None of the evaluation songs or their metadata appear in MF-Skills or in any of our training stages.
>
>
> > I could not find an explanation for why the number of examples increases from 2M songs to 3M examples in MF-Skills. Does this mean captions are added to all 2M songs, and some songs also have QA, resulting in 3M examples?
>
> **Ans.** Exactly. MF-Skills contains ~2.1M unique songs. Every song has a caption, and a subset additionally includes QA pairs. This results in ~3M total training examples (caption samples + QA samples).
> *Additional note:* For the camera-ready version, we will release an expanded dataset with ~3M unique songs and a larger total number of annotations.
>
> > Does AF3-St.3 in Table 2 refer to the training data of the base model Audio Flamingo 3?
>
> **Ans.** Yes. As described in Section 3.1, AF3-St.3 corresponds to our improved Stage-3 fine-tuning of Audio Flamingo 3. This stage enhances the AF3 backbone with additional music-relevant datasets (beyond the original general audio mix) and multi-talker ASR to better handle vocals, duets, and ensemble singing. This strengthened AF3 backbone then serves as the foundation for subsequent Music Flamingo training.
>
> > **Additionally, does "2.0" of MF-SFT mean adding MF-Skills twice?**
>
> **Ans.** No -- “2.0” indicates that MF-Skills is trained for **2 epochs** during MF-SFT (i.e., two full passes over the dataset). This follows the same notation used throughout the Audio Flamingo series. We have clarified this in the caption in the revised paper.
>
> > Are other datasets used in AF3 also reused when fine-tuning Music Flamingo?
>
> **Ans.** Yes. We reuse a subset of music-relevant data from Audio Flamingo 3 Stage-3 during our Stage-4 fine-tuning with MF-Skills. This helps prevent catastrophic forgetting and stabilizes training. Importantly, **only the music-specific portion** of AF3’s training data is reused.
>
> > How were the categories in Figure 27 analyzed?
>
> **Ans.** All captions were passed through **`gpt-oss-120B`**, which generated tags for both *genre* and *cultural origin*. Figure 27 presents the distribution of these tags. For clarity of presentation, we manually merged several closely related or overly fine-grained categories into broader, more interpretable groups.
>
> > **Will the created dataset and source code/checkpoints be made publicly available?**
>
> **Ans.** Yes. A central goal of this work is to advance **open** audio intelligence. We will release all datasets, code, and model checkpoints as part of our open-source effort.
>
> > How were the inputs and outputs for ACC evaluation handled?
>
> **Ans.** For MMAU, MMAU-Pro, MuChoMusic, and MMAR, we follow their *official* MCQ evaluation protocols, which rely on regex-based matching between the model’s answer and the ground truth. For other benchmarks, we use simple string matching to check whether the ground-truth answer appears in the model’s response. This follows the evaluation strategy used in prior work.
>
> > In Figure 3, the term “MF-Skills CoT” is used, but the caption explains it as "MF-Think."
>
> **Ans.** Thank you for catching this. It was a typo -- the correct term is **MF-Think**, and we have fixed this in the revised submission.
>
> > It would be better to standardize the terminology. Figures 4 and 27 appear identical; only Figure 27 is referenced.
>
> **Ans.** Thank you for the suggestion. We have updated the camera-ready to consistently reference **Figure 4**, and its caption now explicitly notes that **Figure 27** provides an enlarged appendix version.
>
> > It would be beneficial to elaborate on how quality filtering is performed.
>
> **Ans.** Thank you for the question. We use Audio Flamingo 3-Think (Stage 3.5) as our verifier. Each generated caption is first decomposed into its constituent aspects (e.g., vocal characteristics, lyrical themes, harmonic structure, mood, production features). The verifier model is prompted to check each aspect independently and produce a short rationale indicating whether the claim is supported by the audio.

---

> > ### Author Response · Authors · 2025-11-19
> > **Response to Official Review of Submission7878 by Reviewer qEKD (2/2)**
> >
> > (continued....)
> >
> > These aspect-level rationales are then passed to `openai/gpt-oss-120b`, which aggregates them and outputs a single "acceptable" / "not acceptable" verdict for a caption using a prompt containing curated human exemplars of both cases. The system tolerates a small amount of noise, but rejects captions that contain incorrect or unverifiable musical claims. This two-stage verification ensures high precision while maintaining good coverage of valid captions.
> >
> > ------
> >
> > ### Weaknesses
> >
> > > The ablation study is missing. Although improvements are made based on AudioFramingo 3, the model is trained in three stages: w/MF-skill, w/MF-think, and GRPO with custom rewards. Having evaluation results for each stage would help confirm each contribution.
> >
> > **Ans.** Thank you for the question. We agree that ablations on the post-training stages are essential for demonstrating the contribution of reinforcement learning. **Our original submission had mentioned results on the GRPO-based RL ablation in lines 455-456.** We had omitted a broader table from the main submission due to space constraints; we did run full ablations. The results below show that (i) removing GRPO leads to clear drops across all reasoning-heavy benchmarks, and (ii) skipping continued AF3 pretraining also degrades performance. These findings underscore that each post-training component -- especially GRPO -- provides meaningful and complementary gains.
> >
> > **Table: Music Flamingo ablation results** (also added as Table 2 in the revised version.)
> >
> > | Ablation               | MMAU  | MMAU-Pro | MuChoMusic |
> > |------------------------|-------|----------|------------|
> > | w/o GRPO               | 75.75 | 63.9     | 69.5       |
> > | w/o cont. AF3 pre.     | 74.27 | 63.0     | 66.2       |
> > | w/o MFSkills           |   73.95  |   61.7      |    47.4     |

---

> ### Author Response · Authors · 2025-11-24
> **Request to review the rebuttal**
>
> Dear reviewer qEKD,
>
> Thank you for taking the time to review our paper. We have addressed your concerns in our submitted response and provided a revised version of the paper. We kindly request you to review our rebuttal and share any additional comments or concerns you may have. Thank you once again for your valuable feedback!
>
> Best,
> Authors of Submission7878

---

> > ### Author Response · Authors · 2025-11-26
> > **Request to review the rebuttal**
> >
> > Dear reviewer qEKD,
> >
> > Thank you for taking the time to review our paper. We have addressed your concerns in our submitted response and provided a revised version of the paper. We kindly request you to review our rebuttal and share any additional comments or concerns you may have. Thank you once again for your valuable feedback!
> >
> > Best,
> > Authors of Submission7878

---

### Official Review · Reviewer_RXYf · 2025-10-28

**Soundness:** 3
**Presentation:** 4
**Contribution:** 2
**Rating:** 6
**Confidence:** 3

**Summary:**

This paper presents Music Flamingo, a large multimodal model extending the Flamingo architecture to handle music, text, and symbolic representations jointly. The model is trained on large-scale music–text–symbol triplets and evaluated across multiple tasks, including music captioning, retrieval, tagging, and question answering. The paper aims to demonstrate the benefits of scaling and unified modeling for general-purpose “music understanding.”

Overall, this is a solid and technically competent piece of work, with careful engineering, broad evaluation, and clear connections to the audio Flamingo. However, several conceptual and empirical gaps remain regarding the necessity of this architecture, its comparison to specialized systems, and its coverage of affective aspects of music understanding.

**Strengths:**

Comprehensive system design — The adaptation of the Flamingo multimodal framework to the music domain is well-motivated and technically executed. Training across multiple modalities (audio, symbolic, text) is a significant engineering contribution.

Unified modeling framework — Unlike prior task-specific MIR models, the same model handles captioning, retrieval, tagging, and QA, demonstrating promising few-shot and zero-shot generalization.

Dataset and reproducibility — The paper provides a substantial and well-curated multimodal dataset and detailed training setup, which is a strength for community reuse.

**Weaknesses:**

1. Necessity over specialized MIR pipelines

While the unified approach is appealing, it remains unclear why such a large, general model is necessary for tasks that existing MIR pipelines already solve effectively (e.g., onset detection, chord recognition, instrument tagging).
The paper could better justify what emergent capabilities arise from this unified model that cannot be achieved by composing established MIR tools.


2. Missing comparisons to domain-specialized music–language models

The paper positions Music Flamingo as a general music–language model but provides limited comparison to existing specialized systems -- I see it cited MuLLaMa but it seems there is no comparison?

3. Conceptual scope and framing

"Music understanding" has cognitive or perceptual components implied by that phrase, not only including facts like pitch, timbre, but also a focus on the feeling and emotional response. This is definitely a hard task and the seems not the focus here. That said, framing the contribution as scaling multimodal representations for music would be more accurate and modest, aligning better with the presented evidence.

**Questions:**

Would you include case studies or analyses showing cross-task generalization or contextual reasoning that would be infeasible for traditional MIR pipelines.

---

> ### Author Response · Authors · 2025-11-19
> **Response to Official Review of Submission7878 by Reviewer RXYf**
>
> We thank you for your thorough review and constructive feedback. We're grateful for your recommendation to accept our paper. We have tried to address each of your concerns point by point.
>
> ------
>
> ### Weaknesses and Questions
>
> > Necessity over specialized MIR pipelines
> > Would you include case studies or analyses showing cross-task generalization or contextual reasoning that would be infeasible for traditional MIR pipelines.
>
> **Ans.** Thank you for the important question. While MIR pipelines excel at isolated, low-level tasks such as onset detection, chord estimation, or instrument tagging, they fundamentally cannot address the broader scope of music understanding targeted in this work. Music Flamingo goes far beyond MIR: as shown in Table 1, it achieves state-of-the-art performance not only on MIR datasets but also on reasoning-centric QA benchmarks (MMAU, MMAU-Pro, MuChoMusic, MMAR), which require multi-step inference, temporal grounding, and interpretation that MIR systems were never designed to handle.
>
> Similarly, Figure 1 highlights that the model’s captions do not merely enumerate MIR-style attributes. They demonstrate layered reasoning: linking tempo/key to harmonic development, tying lyrical content to emotional trajectory, and explaining how specific production choices shape the musical narrative. These holistic descriptions require integrating harmony, structure, vocals, semantics, and cultural context — capabilities that no current MIR tool or MIR-LLM cascade can provide.
>
> The types of questions Music Flamingo answers further illustrate this:
>
> - "In the audio, what action does the singer describe taking despite the pain?"
> - "How do flamenco-style chords influence the melodic identity of the audio?"
> - "When does the E7 chord first clearly appear in the progression?"
> - "Which harmonic progression best differentiates the verse from the chorus?"
> - "Is this music more suitable for romance or mystery?"
>
> (also added in Appendix J of the revised paper uploaded with the rebuttal and on the demo website: https://musicflamingo.github.io/)
>
> These require temporal alignment, harmonic reasoning, lyrical grounding, stylistic interpretation, and high-level semantic judgment — tasks that are not solvable with MIR tools, nor reliably achievable by cascading MIR front-ends with an LLM (which lacks access to the nuanced, time-aligned, multi-level features required for such reasoning).
>
> In short, the unified model is not meant to replace MIR; it is necessary because the capabilities we target — deep, interconnected, theory-aware music understanding -- simply cannot emerge from modular MIR pipelines. Music Flamingo demonstrates these emergent abilities empirically and consistently across the benchmarks.
>
> > Missing comparisons to domain-specialized music–language models
>
> **Ans.**  Thank you for pointing this out, and we apologize for the confusion. As noted in lines 430-440 of the main paper, we do evaluate all major foundational music LLMs, including the ones listed in the Baselines Section 4 -- MuLLaMA included. However, in Table 1 we report only the top-performing prior model for each benchmark to improve readability.
>
> MuLLaMA is not shown there simply because it does not match the performance of the leading baselines on the benchmarks we consider. For completeness, MuLLaMA obtains approximately 29.67 on MMAU, 41.20 on MMAU-Pro, 13.59 on MMAR, and 63.9 on NSynth, which are noticeably below the top open-source and closed-source LALMs shown in Table 1. We will include these numbers in the appendix of the camera-ready version for full transparency.
>
> > "Music understanding" has cognitive or perceptual components implied by that phrase, not only including facts like pitch, timbre, but also a focus on the feeling and emotional response. This is definitely a hard task and the seems not the focus here. That said, framing the contribution as scaling multimodal representations for music would be more accurate and modest, aligning better with the presented evidence.
>
> **Ans.** We agree that full-spectrum "music understanding" encompasses cognitive, perceptual, and affective dimensions, many of which remain open challenges for the field.  *We emphasize that the model goes beyond scaling representations. As shown in Fig. 1 and Table 1, Music Flamingo exhibits reasoning abilities that require connecting musical facts to higher-level interpretations -- e.g., identifying narrative themes in lyrics, explaining how harmonic choices shape mood, or comparing sections based on functional harmony. These tasks explicitly involve aspects of perceptual and semantic understanding that MIR pipelines or representation scaling alone cannot deliver.*
>
> Our intent is not to claim that Music Flamingo fully solves the subjective or emotional aspects of musical perception. Rather, we use "music understanding" in the established ALM/MIR sense: integrating pitch, harmony, structure, lyrics, production, style, and grounded reasoning in a unified model.

---

> > ### Comment · Reviewer_RXYf · 2025-11-27
> >
> > The rebuttal is clear, thorough, and addresses the main points raised in the initial review.
> >
> > The authors have provided convincing evidence that Music Flamingo extends beyond classical MIR pipelines: the unified architecture indeed achieves cross-task generalization and handles reasoning-centric benchmarks. The additional examples in Fig. 1 and Appendix J strengthen the argument.
> >
> > That said, one conceptual question remains: to what extent do the demonstrated abilities truly require end-to-end multimodal integration, rather than being achievable by coupling existing MIR front-ends with a reasoning-capable text model? Many of the illustrated queries—such as chord localization, harmonic contrast, or structural segmentation—could, in principle, be answered by mature MIR tools followed by an LLM that reasons over symbolic features.
> >
> > Where the unified model may provide genuinely new capability is in holistic, perceptual, or stylistic inference, where low-level features interact non-linearly with human affect or cultural context. Examples of reasoning tasks that seem intrinsically hard for a MIR + LLM pipeline (and would highlight the emergent aspect of Music Flamingo) include:
> >
> > 1. Which section of the track is the most emotionally intense?
> > → requires integrating dynamics, harmony, timbre, and lyrical semantics rather than detecting individual cues.
> >
> > 2. Does any segment evoke a jazz-like or East-Asian stylistic color?
> > → involves pattern-level timbral and harmonic analogies that are not explicitly encoded in MIR descriptors.
> >
> > 3. How does production texture (reverb, compression) alter perceived emotional direction?
> >
> > Demonstrating reliable performance on such perceptual or stylistic reasoning would further substantiate the paper’s claim of “deep music understanding.”
> > Overall, the rebuttal strengthens the submission substantially, and I maintain a positive assessment while encouraging the authors to emphasize these remaining frontiers in the final version.
> >
> > Score after rebuttal: No change

---

> ### Author Response · Authors · 2025-11-24
> **Request to review the rebuttal**
>
> Dear reviewer RXYf,
>
> Thank you for taking the time to review our paper. We have addressed your concerns in our submitted response and provided a revised version of the paper. We kindly request you to review our rebuttal and share any additional comments or concerns you may have. Thank you once again for your valuable feedback!
>
> Best,
> Authors of Submission7878

---

> > ### Author Response · Authors · 2025-11-26
> > **Request to review the rebuttal**
> >
> > Dear reviewer RXYf,
> >
> > Thank you for taking the time to review our paper. We have addressed your concerns in our submitted response and provided a revised version of the paper. We kindly request you to review our rebuttal and share any additional comments or concerns you may have. Thank you once again for your valuable feedback!
> >
> > Best,
> > Authors of Submission7878

---

### Official Review · Reviewer_PZ26 · 2025-10-31

**Soundness:** 3
**Presentation:** 2
**Contribution:** 3
**Rating:** 6
**Confidence:** 4

**Summary:**

This paper introduces Music Flamingo (MF), an LLM-based model for music understading tasks. Authors created two large-scale datasets for music understanding tasks: MF-Skills, which contains 4M examples and MF-Think, which contains 300K CoT examples (derived from MF-Skills) for post-training MF to improve its reasoning abilities.  MF-Skills and MF-Think contain full-length music with detailed and multifaceted annotations, compared with previous music understanding datasets that often contain short and surface-level annotations.

Based on the two datasets, authors developed MF by continue training the Audio Flamingo 3 (AF3) model. MF achieved SOTA performances on several public music understanding benchmarks. Human evaluation with music experts show that MF seems better compared with other models.

**Strengths:**

- I think MF-Skills and MF-Think are very good resources for music understanding, where annotations and datasets are rare. The two  datasets could potentially benefit follow-up work of developing better models in this field.

- SOTA performance on several music understanding benchmarks.

- Adding a post-training stage to improve reasoning of the music understanding model is well motivated. Current music understanding   models tend to output surface-level outputs about a music. It is great to see adding post-training RL is also effective for music  understanding.  Authors also show that without post-training, performance drops on MMAU-Pro-Music and MuchoMusic.

**Weaknesses:**

- MF stresses a lot on music with vocal (sec 3.1), and motivates to add several ASR datasets. While this motivation is valid, but non-vocal   music seem less covered in the discussions.

- In the human evaluation from music experts (Tab4), it seems MF often attempts to provide deep details, but often inaccurate. This poses   some concerns on real-world use cases, beyond the good performance on benchmark numbers.

**Questions:**

- Appendix F: how generalizable is this user study? Have you tried with more music/songs, or music without vocals?

**Details Of Ethics Concerns:**

The provided datasets contain music contents, which often involve strict copyright regulations. I think I worth confirming the legal compliance w.r.t. copyrights and license.

---

> ### Author Response · Authors · 2025-11-19
> **Response to Official Review of Submission7878 by Reviewer PZ26 (1/2)**
>
> We thank you for your thorough review and constructive feedback. We're grateful for your recommendation to accept our paper. We have tried to address each of your concerns point by point.
>
> -----
> > MF stresses a lot on music with vocal (sec 3.1), and motivates to add several ASR datasets. While this motivation is valid, but non-vocal music seem less covered in the discussions.
>
> **Ans.** Thank you for pointing this out. We clarify our intent on three fronts:
>
> (1) **Why we emphasize vocal music.** Vocal songs are inherently more complex than instrumental pieces: they combine melody, harmony, timbre, prosody, lyrical meaning, emotional intent, and semantic grounding. Understanding vocals requires solving additional challenges such as lyric alignment, mood and intent inference from vocal delivery, multi-speaker interactions, and disentangling vocals from dense accompaniment. These aspects are precisely the areas where prior LALMs struggle, which motivates our stronger focus on vocal content in Section 3.1.
>
> (2) **Instrumental music is already well-studied in prior work.** Most existing datasets and many LALMs (including earlier Audio Flamingo models) are dominated by instrumental-only clips. Historically, music understanding benchmarks have emphasized tasks such as genre, tempo, chord, and instrument recognition — all of which skew toward instrumental music. In contrast, song understanding (with vocals) has been significantly under-explored, which is why we explicitly highlight it.
>
> (3) **Music Flamingo still covers instrumental music deeply.** Our dataset does include a substantial number of instrumental-only tracks, and Music Flamingo performs strongly on them. For example, the model handles instrumental reasoning tasks in MMAU, MMAU-Pro, and MuChoMusic, and demonstrates detailed structural and harmonic analysis even when no vocals are present. Two representative captioning examples are included below to illustrate this.
>
> ```
> Name: Porter Robinson - Get your Wish (Anamanaguchi Remix)
>
> Description: This track is an aggressive, high‑energy blend of industrial techno and breakcore, marrying the relentless drive of techno's pounding rhythms with the hyper‑fragmented, glitch‑laden aesthetic of breakcore. The duration of the piece is 350.01 seconds. The production is raw and heavily processed, featuring distorted, metallic percussion that is chopped, re‑sequenced, and layered to create a chaotic, ever‑shifting rhythmic landscape. Harsh, abrasive synth textures and noise elements dominate the mix, delivering a dense, abrasive wall of sound that emphasizes texture over melody. The stereo field is deliberately wide and active, with sounds panning rapidly to heighten the sense of disorientation and intensity.
>
> There are no discernible vocals, so the track relies entirely on its instrumental aggression and sound design to convey its mood. The structure is non‑linear and highly dynamic, moving through abrupt transitions, sudden drops, and explosive buildups that keep the listener off‑balance. ... (truncated)
>
> Overall, the piece captures the extreme edge of contemporary electronic music, reflecting the industrial techno and breakcore scenes' fascination with noise, distortion, and the deconstruction of traditional musical forms. Its abrasive aesthetic and kinetic energy place it firmly within the modern underground electronic landscape, appealing to listeners who seek an uncompromising, high‑intensity sonic experience.
> ```
>
> ```
> Name: 【BOFXVI】Angel's Salad【BGA】
>
> This track is an aggressive, high‑energy  Hardcore Techno  piece that leans heavily into  Industrial Techno  aesthetics, blending relentless distorted kicks with harsh, metallic synth textures. The duration of the piece is 178.25 seconds.
>
> Tempo & Key  – The song runs at  120 BPM  and is rooted in  D minor .
>
> Instrumentation & Production  – The foundation is a pounding, heavily distorted kick drum that drives the rhythm, paired with a deep, rumbling sub‑bass that underpins the whole track. Aggressive, metallic synth leads and stabs cut through the mix, while industrial sound‑effects (metallic clangs, filtered noise sweeps, and glitchy samples) add a mechanistic edge. ... (truncated)
>
> Vocals  – No vocal content is present.
>
> Lyrical Themes  – Not applicable (lyrics not provided).
>
> Song Structure & Dynamics  – The arrangement follows a classic techno arc: an atmospheric intro built from filtered noise and subtle percussive clicks, leading into a tension‑building buildup where the distorted kick and synths gradually intensify. .... (truncated)
>
> Theoretical Insight  – Although the harmonic content is sparse, the chord list reveals occasional tonal anchors that punctuate the otherwise rhythm‑focused texture: recurring  Dm7  and  D7  chords provide a minor‑tonic foundation, ... (truncated)
>
> Overall Mood & Context  – The track exudes a dark, chaotic, and intensely mechanical mood, embodying the raw energy of early‑2020s European hardcore ... (truncated)
> ```

---

> ### Author Response · Authors · 2025-11-19
> **Response to Official Review of Submission7878 by Reviewer PZ26 (2/2)**
>
> > In the human evaluation from music experts (Tab4), it seems MF often attempts to provide deep details, but often inaccurate. This poses some concerns on real-world use cases, beyond the good performance on benchmark numbers.
>
> **Ans.** Thank you for the valuable point. We fully agree that real-world usability depends not only on depth but also on accuracy, especially when producing detailed musical analyses. *As mentioned repeatedly in our paper, Music Flamingo is meant to explore music understanding beyond contemporary music benchmarks. For e.g., the depth of music understanding (like the ones shown in Fig. 1) cannot yet be evaluated with current benchmarks.* We further make 3 points that contextualize the expert feedback in Table 4:
>
> (1) The "often inaccurate" comment refers to a single Brazilian example. The evaluator’s remark was specific to one Brazilian folk song -- a musical tradition that is currently under-represented in publicly available datasets and therefore also in our training mix. Brazilian music uses distinctive harmonic colors, regional instruments, and stylistic conventions that the model has had limited exposure to. The example was intentional, meant to transparently illustrate one of the model’s known limitations.
>
> (2) Depth beyond current benchmarks is where Music Flamingo shines, not where it is weak. Across the majority of songs evaluated by experts, the model’s layered descriptions (harmony, structure, vocal timbre, lyrical themes, production details, emotional trajectory) were consistently preferred over existing LALMs. These abilities are not captured by current MIR-style or short-caption benchmarks, which is precisely why we include human expert evaluations.
>
> (3) We view these errors as an important signal, not a flaw. Rather than masking the limitation, we highlight it to motivate the community toward building broader, culturally diverse datasets. Expanding high-quality training data for underrepresented musical traditions is necessary for developing truly global music-understanding models, and Music Flamingo is designed as a step in that direction.
>
> -----
>
> ### Questions
>
> > Questions: Appendix F: how generalizable is this user study? Have you tried with more music/songs, or music without vocals?
>
> **Ans.** Thank you for the question. We agree that assessing generalizability is important. The expert study in Appendix F was limited due to the cost and time involved in detailed, song-level evaluations, but we plan to expand the study for the camera-ready version.
>
> To strengthen the rebuttal, we have already conducted additional analyses on three more songs -- one with vocals and two fully instrumental -- to demonstrate that Music Flamingo’s qualitative advantages extend beyond the initial examples. These additional results have been included in the updated submission and further support that the model’s strengths are not confined to vocal music alone (Appendix G). We plan to add more in the camera-ready!

---

> > ### Author Response · Authors · 2025-11-19
> > **Comment on Ethics Review by Reviewer PZ26**
> >
> > Thank you for raising this important point. We take copyright and licensing considerations very seriously. Below we clarify the compliance measures followed in curating MF-Skills and MF-Think:
> >
> > **(1) We do not distribute copyrighted audio.** Our release contains *only* text annotations (captions, QA pairs, metadata, reasoning chains). We **do not release any copyrighted music audio**, nor do we redistribute any protected media files.
> >
> > **(2) Our work performs text-only analysis; it does not generate or reconstruct music.** Music Flamingo is a **text understanding and reasoning model**, not a generative music system. It does not synthesize or output copyrighted music. All outputs are textual analyses, which fall under typical fair-use research contexts.
> >
> > **(3) All music used for training was sourced from datasets or platforms with permissive reuse conditions.**
> > We rely on:
> > - well-established research datasets containing music cleared for academic use,
> > - publicly available content distributed under **permissive licenses** (e.g., CC-BY, CC-BY-SA, CC-0), or
> > - material whose usage for non-commercial, research-only annotation falls under fair-use provisions.
> >
> > We do **not** collect audio from sources that prohibit research use, redistribution, or derivative annotations.
> >
> > **(4) We do not make any copyrighted content publicly available.** Our open-source release includes **only URLs** (when allowed), metadata, and model-generated text. Users must obtain any audio independently from the original hosting platform under the platform’s terms. No protected audio is stored or redistributed by us.
> >
> > **(5) The project is strictly non-commercial.** All work is conducted for academic research, and the dataset, code, and checkpoints are released under a **research-only license**. No part of the project involves monetization or commercial deployment.
> >
> > In summary, our dataset design and release process explicitly avoid distributing copyrighted audio and rely on text-only annotations derived from ethically sourced material. We believe this aligns with standard research practice and complies with copyright and licensing requirements.

---

> > > ### Comment · Reviewer_PZ26 · 2025-11-24
> > >
> > > Thank you for your reply and the more detailed examples and discussions on the human evaluations -- they addressed my concern. And it is good to confirm that there would not be violations to copyrights etc.
> > >
> > > After reading the response, I would like to maintain my current rating.

---

### Official Review · Reviewer_cqsC · 2025-10-31

**Soundness:** 4
**Presentation:** 3
**Contribution:** 3
**Rating:** 8
**Confidence:** 4

**Summary:**

This work proposes Music Flamingo, a large-scale audio-language model established from a trained Audio Flamingo 3 model, plus 4 post-training steps (3 supervised fine-tuning, 1 RL policy update phase). At the same time, this work proposed MF-Skill and MF-Think, two large-scale synthesized datasets curated for the post-training steps of Music Flamingo. The proposed model achieved top performance on 13 benchmarks (Table 1).

**Strengths:**

- Datasets and model proposed in this work are established on full-length songs rather than clips. This is important as some information can only be well determined by observing the full track (e.g. musical structure, mood, cultural context, etc.).

- Making useful incorporation of speech-centric datasets, such as including multilingual ASR and multi-talker ASR to improve the model capability on bridging texts(e.g. lyrics) and phonemes (speech, vocal).

- The annotations included in MF-Skills incorporated multiple important aspects with not only tags, but also captions and QA pairs. In particular, temporal information is comparatively scarce in public datasets.

- Incorporating a reinforced learning step in the entire pipeline. It is insightful to see a successful adaption of GRPO for LALM.

- Dataset and source code will be published for reproducibility.

**Weaknesses:**

- There's no ablation study on how each post training step improves the model, which is important to show the significance of introducing reinforcement learning. It's also a pity that there's no further discussion about the stability issue of GRPO.

- Although it's a positive sign that proposed work is aware of the dataset issue on cultural bias, according to Figure 4(the same as Figure 27?), the composition of MF-Skills is still western-centric. It can be understood that collecting dataset that covering everything is difficult, however, I believe it is still not sufficient to claim the current composition as "moving from western culture to diverse culture". For example, African, Middle east, India, East asia and Southeast asia music are still under-represented despite they all have huge audience population.

- The reference of GRPO seems to be missing: https://arxiv.org/pdf/2402.03300

**Questions:**

- Regarding quality filtering, what is the frontier model used to verify the captions? Also, what's the model used for re-writing the captions?

- How does the re-writing of MF-Think work? Is that done by repetitive re-generation followed by the verification from the said frontier model?

---

> ### Author Response · Authors · 2025-11-19
> **Response to Official Review of Submission7878 by Reviewer cqsC (1/2)**
>
> We thank you for your thorough review and constructive feedback. We're grateful for your recommendation to accept our paper. We have tried to address each of your concerns point by point.
>
> ------
> ### Weaknesses:
>
> > There's no ablation study on how each post training step improves the model, which is important to show the significance of introducing reinforcement learning.
>
> **Ans.** Thank you for the question. We agree that ablations on the post-training stages are essential for demonstrating the contribution of reinforcement learning. **Our original submission had mentioned results on the GRPO-based RL ablation in lines 455-456.** We had omitted a broader table from the main submission due to space constraints; we did run full ablations. The results below show that (i) removing GRPO leads to clear drops across all reasoning-heavy benchmarks, and (ii) skipping continued AF3 pretraining also degrades performance. These findings underscore that each post-training component -- especially GRPO -- provides meaningful and complementary gains.
>
> **Table: Music Flamingo ablation results** (also added as Table 2 in the revised version.)
>
> | Ablation               | MMAU  | MMAU-Pro | MuChoMusic |
> |------------------------|-------|----------|------------|
> | w/o GRPO               | 75.75 | 63.9     | 69.5       |
> | w/o cont. AF3 pre.     | 74.27 | 63.0     | 66.2       |
> | w/o MFSkills           |   73.95  |   61.7      |    47.4     |
>
> > It's also a pity that there's no further discussion about the stability issue of GRPO.
>
> **Ans.** Thank you for raising this point. GRPO is known to exhibit instability in domains that require heavy exploration or admit many equally valid reasoning paths (e.g., mathematical problem-solving) [1,2]. In contrast, the music-reasoning tasks we target are far less explorative: questions such as "Which line precedes the chorus?" or "When does the E7 chord first appear?" have well-defined, audio-grounded answers.
>
> In our setting, the role of GRPO is not to encourage broad exploration but to elicit structured, faithful reasoning--i.e., to clearly articulate the harmonic, lyrical, and structural cues that lead to a single correct outcome. This substantially reduces the instability typically associated with GRPO, and in practice, we did not observe divergence or oscillatory behavior during training. We will include a brief discussion of this distinction in the revised version in Appendix J.
>
> > lthough it's a positive sign that proposed work is aware of the dataset issue on cultural bias, according to Figure 4(the same as Figure 27?), the composition of MF-Skills is still western-centric. It can be understood that collecting dataset that covering everything is difficult, however, I believe it is still not sufficient to claim the current composition as "moving from western culture to diverse culture". For example, African, Middle east, India, East asia and Southeast asia music are still under-represented despite they all have huge audience population.
>
> **Ans.**  Thank you for the thoughtful observation — we fully agree that achieving genuine cultural breadth in music datasets remains an open challenge. We clarify our position on three fronts:
>
> (1) **Music Flamingo is not a one-off release.** This work represents the first iteration of an ongoing effort. Since the submission deadline, we have already collected ~1M additional captions from underrepresented musical traditions (African, Middle Eastern, Indian, East Asian, Southeast Asian, etc.). These will be included in our open-source release, and we plan to continue expanding the dataset over time.
>
> (2) **The primary bottleneck is annotation, not audio availability.** While acquiring multicultural music audio is feasible, labeling it at the level of detail required by MF-Skills is significantly constrained by current foundational tools—especially multilingual ASR, lyric alignment, and chord/key detection outside Western tonal systems. We view building stronger multilingual and cross-cultural annotation tools as an essential next step, and Music Flamingo is designed to enable exactly this trajectory.
>
> (3) **Even with current limitations, the shift beyond Western music is substantial.** At submission time, over 55% of MF-Skills consisted of non-Western songs, amounting to more than one million culturally diverse tracks. This already exceeds the size and diversity of prior publicly available music datasets (e.g., the Million Song Dataset), and represents a meaningful step toward more inclusive music modeling.
>
> We fully acknowledge that the dataset is not yet culturally complete, but we believe the contributions in this paper -- along with our ongoing expansion efforts -- constitute real progress toward broad, global music understanding.
>
>
> > The reference of GRPO seems to be missing: https://arxiv.org/pdf/2402.03300
>
> **Ans.**  Thank you for pointing this out! We have added the citation to the revised version of the paper uploaded with the rebuttal.

---

> > ### Author Response · Authors · 2025-11-19
> > **Response to Official Review of Submission7878 by Reviewer cqsC (2/2)**
> >
> > ### Questions:
> >
> > > Regarding quality filtering, what is the frontier model used to verify the captions?
> >
> > **Ans.** Thank you for the question. We use Audio Flamingo 3-Think (Stage 3.5) as our frontier verifier. Each generated caption is first decomposed into its constituent aspects (e.g., vocal characteristics, lyrical themes, harmonic structure, mood, production features). The verifier model is prompted to check each aspect independently and produce a short rationale indicating whether the claim is supported by the audio.
> >
> > These aspect-level rationales are then passed to `openai/gpt-oss-120b`, which aggregates them and outputs a single "acceptable" / "not acceptable" verdict for a caption using a prompt containing curated human exemplars of both cases. The system tolerates a small amount of noise, but rejects captions that contain incorrect or unverifiable musical claims. This two-stage verification ensures high precision while maintaining good coverage of valid captions.
> >
> > > Also, what's the model used for re-writing the captions?
> >
> > **Ans.** We use `openai/gpt-oss-120b` for rewriting the captions (and for all text-only rewriting jobs done in the paper). We have added this to the revised version of the paper uploaded with the rebuttal.
> >
> > > Also, what's the model used for re-writing the captions?
> >
> > **Ans.** Thank you for the question. All text-only rewriting in the paper -- including MF-Think -- is performed using `openai/gpt-oss-120b`, and we have clarified this in the revised submission.
> >
> > For MF-Think specifically, the rewriting process is not repetitive regeneration. Instead, it is a targeted, aspect-level correction pipeline:
> >
> > **Verification step:** Our verifier model inspects each step of the reasoning chain. For any step it flags as incorrect (“No”), we isolate only that specific aspect.
> >
> > **Template-based rewrite prompting:** We then construct a structured prompt that includes: the full original chain-of-thought, the specific aspect identified as incorrect, and the relevant ground-truth metadata from MF-Skills.
> >
> > **Selective rewriting:** `openai/gpt-oss-120b` is instructed to rewrite only the incorrect portion, while keeping all other (verified) reasoning steps unchanged.
> >
> > Below is an illustrative example (abbreviated in the rebuttal):
> >
> > Original reasoning: “From the first verse into the chorus, I hear bright acoustic strums layered with crunchy electrics …”
> >
> > Verifier check: “Do you hear bright acoustic strums layered with crunchy electrics?” --> No
> >
> > Rewrite instruction: The model is told: "You are provided with the CoT for a QA from a music reasoning dataset. Additionally, you are provided with a pointer to a wrongful step and the rationale behind it. FInally, you are also provided with the ground-truth metadata about the song. Rewrite only this part while preserving all other correct steps."
> >
> > *This ensures that MF-Think contains faithful, metadata-grounded reasoning chains, without introducing drift or unnecessary rewrites.*
> >
> >
> > ### References
> >
> > [1] Zheng, Chujie, et al. "Group sequence policy optimization." arXiv preprint arXiv:2507.18071 (2025).
> > [2] Dai, Muzhi, Shixuan Liu, and Qingyi Si. "Stable Reinforcement Learning for Efficient Reasoning." arXiv preprint arXiv:2505.18086 (2025).

---

> ### Author Response · Authors · 2025-11-24
> **Request to review the rebuttal**
>
> Dear reviewer cqsC,
>
> Thank you for taking the time to review our paper. We have addressed your concerns in our submitted response and provided a revised version of the paper. We kindly request you to review our rebuttal and share any additional comments or concerns you may have. Thank you once again for your valuable feedback!
>
> Best,
> Authors of Submission7878

---

> > ### Author Response · Authors · 2025-11-26
> > **Request to review the rebuttal**
> >
> > Dear reviewer cqsC,
> >
> > Thank you for taking the time to review our paper. We have addressed your concerns in our submitted response and provided a revised version of the paper. We kindly request you to review our rebuttal and share any additional comments or concerns you may have. Thank you once again for your valuable feedback!
> >
> > Best,
> > Authors of Submission7878

---

### Meta-Review · Area_Chair_uVK2 · 2026-01-07

**Summary:**

The reviewer consensus finds that this is a well-executed paper that significantly advances multi-modal music modeling (the Music Flamingo model) as well as resources for the community (MF-Skills and MF-Think datasets). Concerns raised by reviewers are constructive and philosophical rather than dealbreakers: large unified models vs specialized MIR pipelines, the completeness of ablation studies, cultural coverage, framing of “music understanding” versus multimodal representation scaling.

**Reviewer Concerns:**

Reviewer concerns are mostly about thoroughness and framing rather technical soundness. Reviewers asked for clearer ablations of the postraining pipeline (MF-Think, continued AF3 training, GRPO). They also critiqued dataset coverage and documentation (cultural diversity balance, vocal vs. instrumental emphasis, collection/filters/leakage). And there were some bigger-picture questions about whether a large unified model is necessary vs. MIR feature extraction + a reasoning LLM, and whether “music understanding” overclaims the experimental results.

The rebuttal addresses several substantive concerns: ablations to isolate aspects of the post-training pipeline, GRPO stability, dataset construction and filtering details, copyright and licensing, and questions about vocal vs. instrumental coverage. Some questions remain about the necessity of a unified Music Flamingo architecture vs. MIR feature extraction + LLM reasoning,

**Reviewer Scores:**

Reviewers were generally positive before discussion. Scores wouldn't have moved down and the paper is already within accept territory.

---

### Decision · Program_Chairs · 2026-01-26

Accept (Poster)